# *STK11* (LKB1) missense somatic mutant isoforms promote tumor growth, motility and inflammation

Paula Granado-Martínez [1,11], Sara Garcia-Ortega [1,11], Elena González-Sánchez[1,11], Kimberley McGrail [1], Rafael Selgas [1], Judit Grueso[1,2], Rosa Gil[1], Neia Naldaiz-Gastesi[1,9], Ana C. Rhodes[1,10], Javier Hernandez-Losa[3], Berta Ferrer[1,3], Francesc Canals[4], Josep Villanueva[5], Olga Méndez[5], Sergio Espinosa-Gil[6], José M. Lizcano [6], Eva Muñoz-Couselo[1,7], Vicenç García-Patos [1,8] & Juan A. Recio [1✉]

Elucidating the contribution of somatic mutations to cancer is essential for personalized medicine. *STK11* (*LKB1*) appears to be inactivated in human cancer. However, somatic missense mutations also occur, and the role/s of these alterations to this disease remain unknown. Here, we investigated the contribution of four missense LKB1 somatic mutations in tumor biology. Three out of the four mutants lost their tumor suppressor capabilities and showed deficient kinase activity. The remaining mutant retained the enzymatic activity of wild type LKB1, but induced increased cell motility. Mechanistically, LKB1 mutants resulted in differential gene expression of genes encoding vesicle trafficking regulating molecules, adhesion molecules and cytokines. The differentially regulated genes correlated with protein networks identified through comparative secretome analysis. Notably, three mutant isoforms promoted tumor growth, and one induced inflammation-like features together with dysregulated levels of cytokines. These findings uncover oncogenic roles of LKB1 somatic mutations, and will aid in further understanding their contributions to cancer development and progression.

[1] Biomedical Research in Melanoma-Animal Models and Cancer Laboratory- Vall d'Hebron Research Institute VHIR-Vall d'Hebron Hospital Barcelona-UAB, Barcelona 08035, Spain. [2] Experimental Therapeutics Group, Vall d'Hebron Institute of Oncology (VHIO), Barcelona 08035, Spain. [3] Anatomy Pathology Department, Vall d'Hebron Hospital Barcelona-UAB, Barcelona 08035, Spain. [4] Proteomics Laboratory, Vall d'Hebron Institute of Oncology (VHIO, Barcelona 08035, Spain. [5] Preclinical Research Program, Vall d'Hebron Institute of Oncology (VHIO, Barcelona 08035, Spain. [6] Protein Kinases and Signal Transduction Laboratory, Neuroscience Institute and Molecular Biology and Biochemistry Department, UAB, Bellaterra, Barcelona 08193, Spain. [7] Clinical Oncology Program, Vall d'Hebron Institute of Oncology (VHIO), Vall d'Hebron Hospital, Barcelona-UAB, Barcelona 08035, Spain. [8] Dermatology Department, Vall d'Hebron Hospital Barcelona-UAB, Barcelona 08035, Spain. [9] Present address: Biodonostia, Neurosciences Area, Group of Neuromuscular Diseases, San Sebastian 20014, Spain. [10] Present address: Barcelona Clinic Liver Cancer (BCLC) Group, Liver Unit, Hospital Clínic of Barcelona, University of Barcelona, Institut d'Investigacions Biomèdiques August Pi i Sunyer (IDIBAPS), Barcelona 08036, Spain. [11] These authors contributed equally: Paula Granado-Martínez, Sara García Ortega, Elena González-Sánchez. ✉email: juan.recio@vhir.org

STK11 (Liver kinase 1, LKB1) was first identified as a tumor suppressor gene through its association with the Peutz-Jeghers Syndrome (PJS)[1]. STK11 appears to be inactivated or mutated in sporadic cancers whose spectrum of tumor types suggests cooperation with exposure to environmental carcinogens. Thus, alterations in LKB1 have been found in non-small-cell lung cancer (NSCLC), malignant melanoma, and cervical cancer among others[2–4].

The serine/threonine kinase LKB1 belongs to the calcium calmodulin family, which is ubiquitously expressed in several tissues and highly conserved among eukaryotes. Over the past 15 years, LKB1 has been implicated in a number of essential biological processes such as: cell cycle control[5,6], cellular energy metabolism[7,8], angiogenesis[9,10], cell polarity[11], and DNA damage response[12]. The sub-cellular localization and activity of LKB1 is controlled through its interaction with STRAD and the armadillo repeat-containing mouse protein 25 (Mo25)[13,14]. LKB1 regulates the activity of at least 14 downstream kinases related to the AMPK family[15] and phosphorylates other substrates including STRAD[16], PTEN[17], and p21CDKN1A[12]. LKB1 is phosphorylated on at least eight residues, and evidence suggests that LKB1 autophosphorylates itself on at least four of these, whereas the other four are phosphorylated by upstream kinases[8,16]. While these post-translational modifications seem not to modify its kinase activity, they are involved in the different biological responses associated with LKB1, and likely in its interactions with other partners.

Up to date, more than 400 unique mutations have been described for the STK11 gene, where ~70% of these mutations promote the truncation of the protein and the other 30% represent missense mutations (COSMIC and TCGA-Bioportal). As a tumor suppressor, a number of studies have shown the contributions of the genetic loss of LKB1 to tumorigenesis. It has been demonstrated that LKB1 controls cell cycle through the transcriptional regulation of Cyclin D1 and p21CDKN1A[56], where re-expression of LKB1 leads to G1 cell cycle arrest. The role of LKB1 in controlling cell metabolism through AMPK signaling has been widely documented. We know that the LKB1-AMPK axis controls lipid and glucose metabolism, and acts as a negative regulator of the Warburg effect suppressing tumor growth[8,18]. LKB1 is also important in the regulation of catabolic pathways leading to the increase of glucose uptake and modulation of glycolysis[19] or the mobilization of lipid stores by stimulating lipases, such as adipose triglyceride lipase, to release fatty acids from triglyceride stores[20]. LKB1-AMPK-stimulated pathways also include increased turnover of macromolecules by autophagy, allowing the turnover of old and damaged molecules, or the replenishment of nutrient stores under starvation[20]. Additionally, several investigations have suggested the role of LKB1 in regulation of physiological[21] and pathological angiogenesis[22] through the regulation of VEGF, MMP-2, MMP-9, bFGF, and NOX1 expression, and its participation in neurophilin-1 degradation[23–25]. Studies of LKB1 loss of function have also revealed its role in cell polarity and motility through the regulation of PAK1[15] and the modulation of the phosphorylation status of FAK and CDC42 activation[26]. Together, these functions contribute to the induction of epithelial mesenchymal transition (EMT) and metastasis[27,28]. In addition to this, in vivo experiments have shown evidence for the contribution of LKB1 to genotoxic DNA damage response and DNA damage repair[12,29].

Despite our increased knowledge about contributions of the loss of LKB1 in the different biological responses, much less is known about the specific biological contributions of the different STK11 mutants identified in human tumors. Previous studies have elucidated the tumor suppressor capabilities and the kinase activity of several mutants found in a group of PJS patients. Here we show the functional consequences of the expression of LKB1Y49D, LKB1R87K, LKB1G135R, and LKB1D194Y human tumor missense mutants compared to the wild-type (LKB1WT) isoform. These mutants were selected according to their location in the 3D LKB1 structure, which affect different parts of the protein and may contribute to its different functions. We investigated their tumor suppressor effectiveness and their contributions to cell cycle regulation, motility, and modification of the extracellular microenvironment. Finally, we also investigated their contribution to in vivo tumor growth.

## Results

### LKB1Y49D, LKB1G135R, and LKB1D194Y mutants lack LKB1WT tumor suppressor activity.

Initially, mutated residues were localized within the primary and 3D LKB1 protein structure (heterotrimeric LKB1-STRADα-MO25α complex (2WTK.pdb)). While the Y49 residue was embedded within a β-sheet in the LKB1 N-lobe, two consecutive arginines R86 and R87 in the αB-helix were completely exposed on the surface of the protein, and the G135 and D194 residues were located at the ATP-binding cleft of LKB1 (Fig. 1a and Supplementary Fig. 1a), suggesting different functional consequences for these mutations. To study the biological contributions of LKB1 mutant isoforms to tumor cells, we initially infected A549 cells, which lack the expression of LKB1, with an inducible lentiviral vector containing the different LKB1 isoforms allowing the expression of physiological amounts of LKB1 (Fig. 2b, c and Supplementary Fig. 1b, c). The result of colony-formation assays and proliferation curves confirmed the tumor-suppressive activity of LKB1WT. LKB1R87K conserved the tumor suppression activity of LKB1WT, however, LKB1Y49D, LKB1G135R, and LKB1D194Y isoforms lost this function (Fig. 1d, e). These results were corroborated in HeLa and G361 cell lines, which were also null for LKB1 expression (Supplementary Fig. 1d, e). Furthermore, these data were supported by cell cycle analysis studies in which expression of LKB1WT and LKB1R87K but not the other isoforms led cells to be arrested in G1 phase (Fig. 1f). Metabolic profiling of cells showed a significant increase in mitochondrial proton leak in LKB1G135R expressing cells that might have a have a major impact on mitochondrial coupling efficiency. In relation to this, G135R expressing cells also showed a significant increase of ECAR. Although not significant, D194Y expressing cells also showed a higher acidification rate, suggesting an increase use of glycolysis by cells expressing any of these two mutants (Fig. 1g). Mitochondrial dysfunction in LKB1G135R expressing cells correlated with their resistance to metformin, which is known to inhibit mitochondrial respiration (Supplementary Fig. 1f). Thus, although LKB1R87K conserves in vitro tumor suppression capabilities, LKB1Y49D, LKB1G135R, and LKB1D194Y have lost this functional feature.

### Kinase activity, localization, and stability of LKB1 mutant isoforms.

Since the mutations affect different structural domains of LKB1, we next investigated the specific kinase activity of the different LKB1 mutant isoforms compared to that of wild type. In vitro kinase assays in the presence of STRADα showed that the LKB1R87K mutant exhibited similar kinase activity to the wild-type isoform. Although LKB1D194A has been described as a kinase-dead mutant[6], LKB1D194Y and LKB1Y49D showed ~30% of the LKB1WT activity ($p < 0.001$), and LKB1G135R showed a 40% decrease in the kinase activity, compared to the wild-type isoform (Fig. 2a).

LKB1 localizes to the nucleus through its nuclear localization signal (NLS) and it is translocated to the cytoplasm upon binding to STRADα[13], which is also essential to fully activate LKB1 kinase activity. Immunofluorescence experiments showed nuclear and

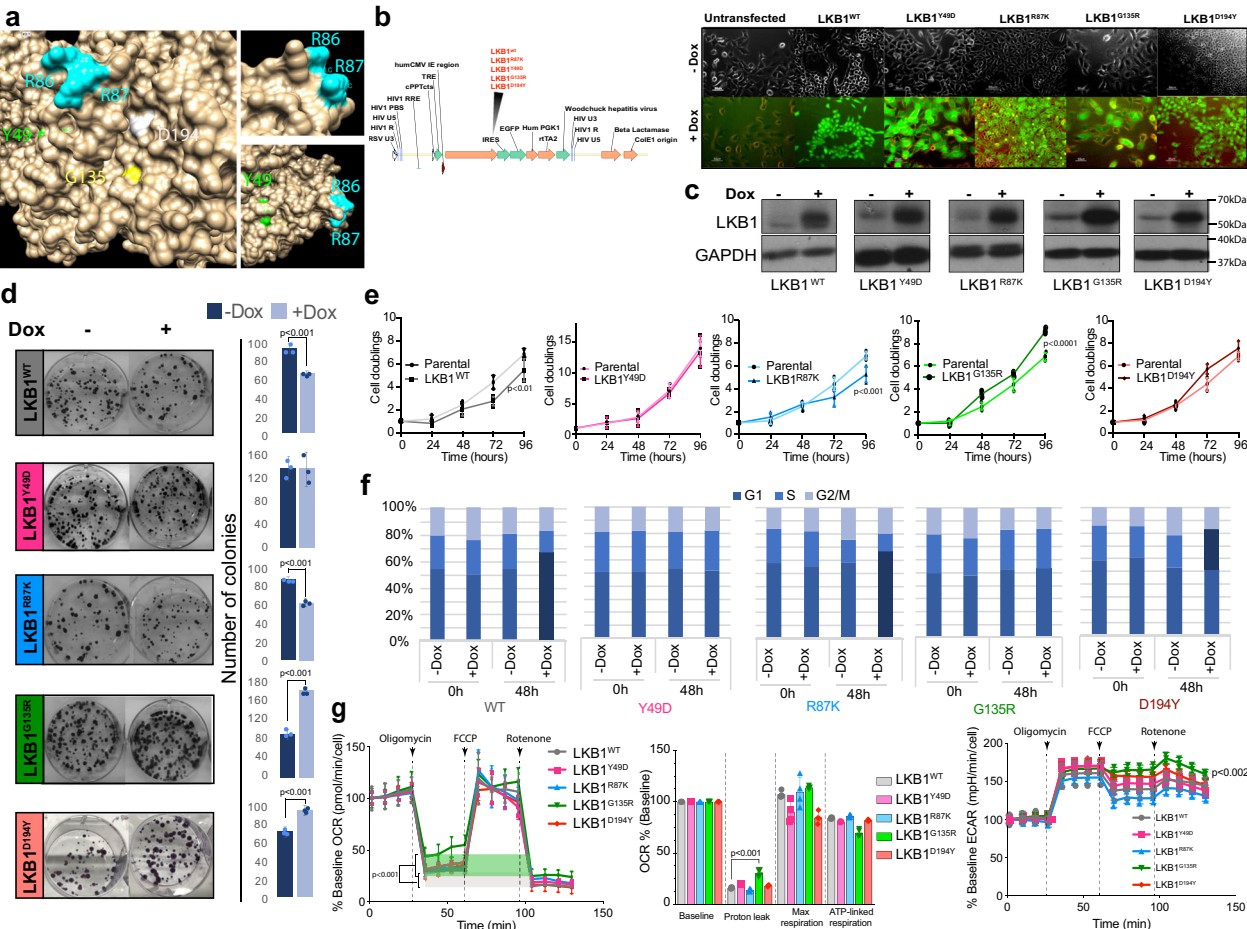

**Fig. 1 LKB1$^{Y49D}$, LKB1$^{G135R}$, and LKB1$^{D194Y}$ mutants lack the LKB1$^{WT}$ tumor suppressor activity. a** Localization of mutated residues within the 3D structure of the LKB1-STRADα-MO25α complex (2WTK.pdb). Mutated residues are indicated. **b** Vector map used to clone the different LKB1 isoforms; on the right, expression of GFP in uninduced and induced infected cells with the different constructs is shown. **c** Western blot, showing the induction of expression of the different LKB1 isoforms in A549 cells, 24 h after doxycycline treatment. **d** Clonogenic assay with cells expressing the different LKB1 isoforms. Graphs show the quantification of the number of colonies. ($n = 3$ experiments ±SD). **e** Representative proliferation curves, showing the effect of LKB1 mutants on cell proliferation ($n = 3$ experiments ±SD; $p$-value was calculated by Student's $t$ test). **f** Graphs showing cell cycle analysis of cells expressing the different LKB1 isoforms for 48 h. Dark blue bars represent cell cycle phases showing significant changes ($n = 3$, $p < 0.05$ calculated by Student's $t$ test). **g** Graphs showing oxygen consumption rates (OCR), parameters associated (oxygen consumption baseline; mitochondria proton leak; mitochondrial maximal respiration and ATP-linked respiration), and the extracellular acidification rates (ECAR). The green area and the gray area represent proton leak from LKB1$^{G135R}$ and LKB1$^{WT}$ expressing cells, respectively ($n = 5$ ± SD; $p$-value was calculated by Student's $t$ test).

cytoplasmic localization of LKB1$^{WT}$, LKB1$^{R87K}$, LKB1$^{G135R}$, and LKB1$^{D194Y}$. However, besides to the presence of STRADα, LKB1$^{Y49D}$ showed a mainly nuclear localization in the three different cell lines tumor models (A549, HeLa, and G361), suggesting a lack of binding of this isoform to STRADα (Fig. 2b and Supplementary Fig. 2a, b). Indeed, analysis of the LKB1 immunocomplexes in two different cell lines (A459 and G361) confirmed the lack of binding of LKB1$^{Y49D}$ to STRADα, which is complimentary to the diminished kinase activity displayed by this mutant (Fig. 2c and Supplementary Fig. 2c). The compromised kinase activity of LKB1$^{Y49D}$, LKB1$^{G135R}$, and LKB1$^{D194Y}$ mutants was confirmed through its direct target AMPK upon stress metabolic conditions (Fig. 2d). In addition, LKB1$^{R87K}$, LKB1$^{Y49D}$, and LKB1$^{D194Y}$ isoforms showed a significant decrease in the half-life of the protein compared to LKB1$^{WT}$ (Fig. 2e). Thus, these data show that the investigated LKB1 mutant isoforms have diminished kinase activity through either different mechanisms or decreased protein stability. These data are also in agreement with the tumor suppressor role of the protein and the selection of these mutations in human cancer.

**LKB1$^{R87K}$ and LKB1$^{D194Y}$ confer tumor cells with increased in vitro motility**. It has been described that LKB1 plays a role in metastasis, adhesion, and motility[30]. Furthermore, LKB1 serves as a focal adhesion kinase (FAK) repressor, where LKB1 depletion causes rapid focal adhesion site turnover[31]. Thus, we investigated the specific contribution of the different isoforms to cell motility. As described previously, LKB1$^{WT}$ expression repressed FAK$^{Y397}$ and FAK$^{Y861}$ phosphorylation. However, LKB1$^{R87K}$, LKB1$^{G135R}$, or LKB1$^{D194Y}$ neither repressed FAK activation nor increased the phosphorylation of these residues (Fig. 3a), including the FAK$^{Y576}$ residue, which is the target of c-SRC (Fig. 3a), suggesting a possible alteration of motility by LKB1 mutants. Taking advantage of the inducible lentiviral construct, we measured the distance traveled by cells upon expression of the different mutants by tracking the distance migrated by fluorescent cells upon doxycycline treatment. LKB1$^{R87K}$-expressing cells showed significantly increased motility compared to wild-type cells or cells expressing the other mutant isoforms. The LKB1$^{D194Y}$ mutant also promoted cell motility albeit not as strongly as LKB1$^{R87K}$ (Fig. 3b).

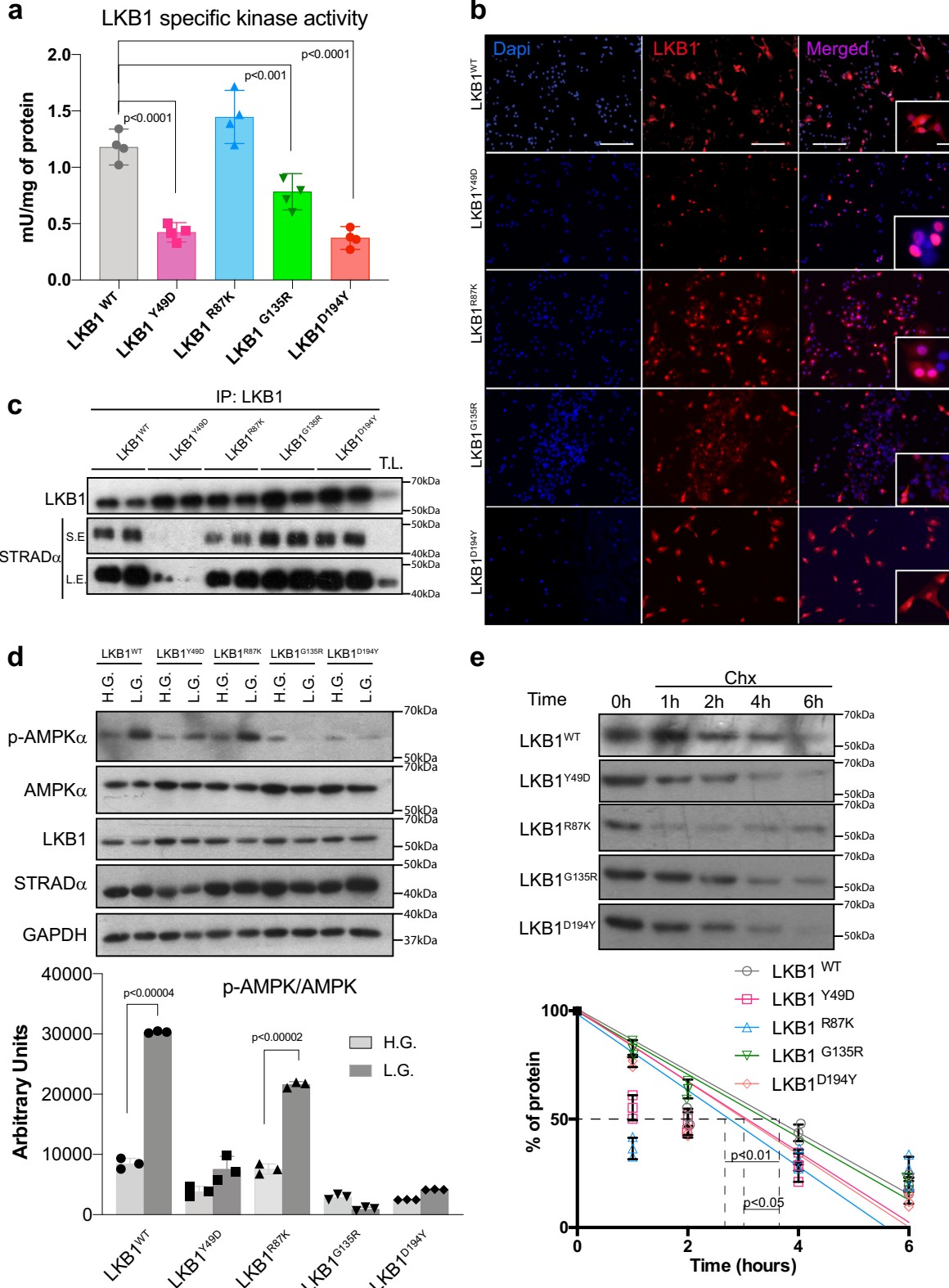

Moreover, expression of LKB1$^{R87K}$ but not LKB1$^{WT}$ induced the formation and reorganization of F-actin fibers (Fig. 3c). Matrigel 3D cultures showed that expression of LKB1 promoted a significant increase in the number of spheres generated; however, only the expression of the LKB1$^{Y49D}$, LKB1$^{G135R}$, and LKB1$^{D194Y}$ mutant isoforms increased the size of the spheres (Fig. 3d). As expected, expression of LKB1 promoted β-catenin

degradation[32], which was more evident upon LKB1$^{R87K}$ or LKB1$^{D194Y}$ expression (Fig. 3e). Loss of β-catenin at the cell junctions upon expression of the LKB1$^{R87K}$ or LKB1$^{D194Y}$ mutants was also observed in the 3D cultures (Fig. 3f). Altogether, the data suggest that LKB1$^{R87K}$ and to a lesser extent LKB1$^{D194Y}$, induce cell motility, promoting cytoskeleton regulation and β-catenin degradation.

**Fig. 2 Kinase activity, subcellular localization, and stability of LKB1 mutant isoforms. a** Graph showing the specific in vitro kinase activity of LKB1 mutant isoforms ($n = 4$ assays per isoform ±SD; $p$-values were calculated by Student's $t$ test). **b** Representative immunofluorescence images showing the subcellular localization of LKB1 isoforms in A549 cells. DAPI staining shows nuclear staining (bars represent 250 μm and 50 μm respectively). **c** Western blot showing the amount of STRADα bound to LKB1 isoforms. Duplicate of immunoprecipitated LKB1 complexes are shown for each isoform in A549 cells. T.L. total lysate, L.E. long exposure, S.E. sort exposure. **d** Western blot showing the amounts of p-AMPK upon glucose starvation (low glucose (L.G.)) or complete media (high glucose (H.G.)). The graph shows quantification of p-AMPK ($n = 3$ experiments ±SD). LKB1 show the amounts of mutant isoforms. AMPK and GAPDH are showed as a loading control. **e** LKB1-isoform protein stability. A549 cells were treated with cycloheximide. (CHX, 5 μg/mL). Then a pulse chase experiment was performed for the indicated time points. The graph shows quantification of LKB1 isoforms ($n = 3$ experiments ±SD). A representative western blot is shown on the right. Dashed lines show the half-life of proteins. The $p$-value was calculated by Student's $t$ test.

**Vesicle trafficking regulatory molecules, adhesion molecules, and cytokines are differentially regulated by LKB1 mutant isoforms.** To gain knowledge about the molecular mechanisms involved in the observed phenotypes, we initially compared the gene expression profile of A459 cells expressing and not expressing LKB1$^{WT}$. Unsupervised hierarchical clustering of genes (817 genes 1.2-fold and $p > 0.05$) showed that LKB1$^{WT}$ promoted a distinct gene expression profile. To strength the relevance of our gene dataset, we compared our 817 regulated genes with the 2080 unique regulated genes obtained from 15 different datasets (top 200 regulated genes in each data set) published in Kaufman et al.[33], comparing the gene expression profiles of human and murine tumors with or without the genetic loss of *STK11*. One hundred and fifty-one genes were common to both lists, where 25% of the genes were also associated to the functional clusters described in that study[33] (Supplementary Fig. 3a). Gene set enrichment analysis showed that morphogenesis and development, cell signaling, proliferation, adhesion and motility, metabolic regulation, and immunomodulation were among the processes most significantly affected by LKB1$^{WT}$ expression (Fig. 4a). The most significantly regulated genes (FDR < 0.1) included *STK11*, *AP1S3*, and *RUSC2* (upregulation), which are involved in intracellular vesicle trafficking and *PLAUR*, and *TGFB1I1*, which regulate cell migration, cell growth, proliferation, and the transcriptional activity of nuclear receptors. LKB1 also promoted the downregulation of important immunomodulators and cytokines such as: *SEMA7A*, IL8, *CXCL1*, metabolic enzymes (*TDO2* and *SOD2*), and adhesion molecules such as *CDH1* (Fig. 4b). The regulation of these genes upon LKB1$^{WT}$ reintroduction was validated by RT-PCR in a subset of different samples confirming the regulation of these processes (Fig. 4c). Next, we asked whether the expression of these genes was differentially regulated by the LKB1 mutant isoforms compared to LKB1$^{WT}$. All four mutants showed defective regulation of genes involved in intracellular trafficking of vesicles and endosomes (*AP1S3* and *RUSC2*) and of *CDCP1*, a molecule involved in cell adhesion and modulated by Src kinases (Fig. 4d). *TGFB1I1*, a molecular adapter linking various intracellular signaling modules to plasma membrane receptors regulating the Wnt and TGFβ signaling pathways, was also downregulated by LKB1$^{R87K}$ or LKB1$^{D194Y}$. *PLAUR* a protein involved in processes related to cell-surface plasminogen activation and localized degradation of the extracellular matrix was upregulated in LKB1$^{R87K}$ or LKB1$^{G135R}$, and *DUSP1* which is associated to the negative regulation of cellular proliferation was downregulated in LKB1$^{Y49D}$, LKB1$^{G135R}$, and LKB1$^{D194Y}$-expressing cells (Fig. 4d). One interesting observation was the downregulation of *IL8* and *CXCL1* upon LKB1$^{WT}$ expression, whereas expression of LKB1$^{Y49D}$ and LKB1$^{D194Y}$ isoforms either increased or failed to downregulate the expression of these immunomodulators (Fig. 4d). *TDO2*, a rate-limiting enzyme in tryptophan metabolism, was differentially regulated in LKB1$^{Y49D}$, LKB1$^{G135R}$, and LKB1$^{D194Y}$-expressing cells, while the antiapoptotic and stress protective protein *SOD2* was markedly upregulated by LKB1$^{D194Y}$. *CYP1B1* a detoxification protein

belonging to the cytochrome p450 family is dysregulated in *LKB1$^{R87K}$* and *LKB1$^{G135R}$* mutant cells (Fig. 4d). Albeit the low number of *LKB1$^{D194}$* mutated samples in the Lung adenocarcinoma Pan Cancer dataset (TCGA), the expression of most of the genes above investigated correlated with the observed in the mutated human samples (Supplementary Fig. 3b). Altogether, these data show evidence supporting the aberrant regulation by LKB1 mutant isoforms of several processes contributing to tumor development and progression, including intracellular vesicle trafficking, cell adhesion and motility, immunomodulation and metabolism.

**Analysis of LKB1-isoform secretomes identifies differential alterations in exocytosis regulation, cell adhesion and motility, and cytokine signaling.** Since LKB1 expression was shown to regulate vesicle trafficking, cell adhesion and motility, and cytokine signaling, we decided to analyze the secretome of cells expressing the different LKB1 isoforms to validate and confirm some of the gene expression results and observed phenotypes. Relative label-free protein quantification analysis from both uninduced and induced cell secretomes expressing each LKB1 isoform allowed us to detect 594, 640, 617, 681, and 596 differentially secreted proteins from cells expressing LKB1$^{WT}$, LKB1$^{Y49D}$, LKB1$^{R87K}$, LKB1$^{G135R}$, or LKB1$^{D194Y}$ expressing cells, respectively (Supplementary Fig. 4a and Supplementary Data 1 and 2). In the case of LKB1$^{WT}$, six proteins identified in the secretome (BMP1 VCL, SOD2, IL8, GREM1 and CXCL1) were also present in the top 31 regulated genes (FDR < 0.1) upon LKB1$^{WT}$ expression and were regulated in the same direction (Supplementary Fig. 4b). Overall, 60–75% percent of the differentially expressed proteins (440 proteins) upon LKB1 expression were common to all isoforms, and only 1–12% of the proteins were identified upon specific LKB1-isoform expression (Fig. 5a and Supplementary Data 3). Pathway and biological process enrichment analysis of the list of proteins regulated by all LKB1 isoforms (Metascape.org) showed that exocytosis regulation, extracellular matrix organization and remodeling, wound-healing response, signaling by interleukins and response to oxidative stress were the most significantly enriched processes. (Fig. 5a and Supplementary Fig. 5). Most of the proteins that were upregulated in the secretome of cells expressing the different LKB1 mutants compared to LKB1$^{WT}$ were either common to the different protein lists (Fig. 5b circos plot purple lines) or proteins that belonged to the same enriched ontology term (Fig. 5b circos plot blue lines). Again, biological process enrichment analysis of all upregulated proteins from these four groups (upregulated proteins in LKB1 mutants vs. LKB1$^{WT}$) resulted in the identification of processes such as protein processing, cell adhesion and motility metabolic regulation and cytokine signaling (Fig. 5b). The results from the individual analysis of each LKB1 mutant compared to LKB1$^{WT}$ agreed and supported previous observations. The LKB1$^{Y49D}$ mutant induced the secretion of proangiogenic, inflammatory and immunomodulator molecules (CXCL1, CXCL2, CXCL3, CXCL5, IL8 MMP7 and TIMPs) (Fig. 5c). In

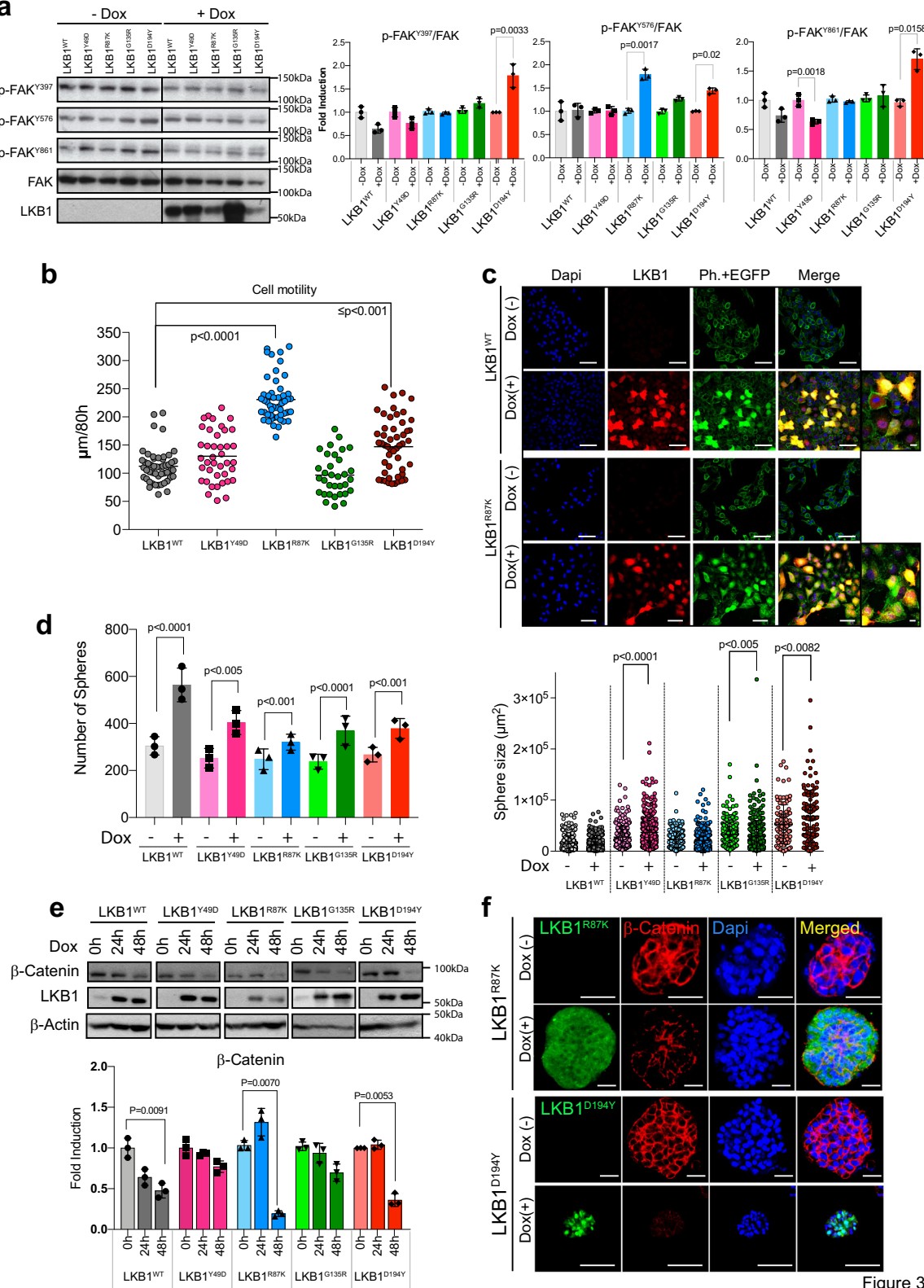

Figure 3

agreement with the above results, LKB1[R87K] promoted the secretion of molecules related to extracellular matrix remodeling, cytoskeletal maintenance and adhesion, such as MMP2, FN1, IQGAP1 and FLNA (Fig. 5d). Expression of LKB1[G135R] and LKB1[D194Y] promoted the alteration of common networks of proteins, including protein-processing components (PSME1, PSME2 and PSME3), as well as the increased secretion of

molecules related to extracellular remodeling and adhesion (VCNA, VTN, FLNA, VIM and IQGAP1) (Fig. 5e-f). In the case of LKB1[D194Y], we also identified an increase in the number of molecules involved in the activation of growth factors (cathepsins and PAPPA), mitogenic growth factors (PDGFD) and immuno-modulators (IL8) (Fig. 5f). Thus, these results support the described phenotypic and molecular observations related to each

**Fig. 3 Differentially regulated cell motility by LKB1 mutant isoforms. a** Western blot showing the regulation of FAK phosphorylation after the expression of LKB1 mutant isoforms for 48 h. Graphs show quantification of every phosphorylation site. ($n = 3$, $p$-value was calculated by Student's $t$ test). **b** A graph showing the quantification of A549 cell motility induced by LKB1 isoforms. The movement of 35–50 cells expressing each different isoform (colored circles) was quantified for 80 h and plotted. The $p$-value was calculated by Student's $t$ test. **c** Representative images of LKB1 immunofluorescence (red) and F-actin (phalloidin, green). Cells appear filled with green upon doxycycline treatment because of IRES-GFP expression linked to LKB1 expression. Bars represent 100 μm and 20 μm. **d** Graphs showing the quantification of the number and size of spheres grown in Matrigel upon expression of the indicated LKB1 isoform. For number of spheres experiments were performed in triplicate. For determining size, 200 spheres per group (from the three experiments) were analyzed. $p$-value was calculated by Student's $t$ test. **e** Representative western blot showing the amount of β-Catenin upon LKB1 isoform induction. β-Actin is shown as loading control. Graphs shows normalized quantification. ($n = 3$, $p$-value was calculated by Student's $t$ test). **f** Immunoflurescence of Matrigel spheres showing the amounts of β-catenin upon the expression of the indicated LKB1 isoforms. DAPI staining shows nuclear staining. Bars represent 500 μm.

LKB1 mutant in regulating cell proliferation, cell motility and cytokine production.

**Distinct contributions of LKB1 mutant isoforms to *in vivo* tumor growth.** Next, we studied the specific contribution/s of each LKB1 mutant to in vivo tumor growth. As expected (Fig. 1), LKB1$^{WT}$ and LKB1$^{R87K}$ behaved as tumor suppressors. The reduction in tumor growth was more evident for the wild-type isoform, and in both cases (LKB1$^{WT}$ and LKB1$^{R87K}$), some of the tumors showed more differentiated areas with no proliferating (Ki67-negative) nested polarized epithelial cells (Fig. 6a, b). LKB1$^{Y49D}$ promoted tumor growth and, in agreement with the above results, these tumors presented signs of hemorrhage and inflammation that actively contributed to the tumor size (Fig. 6c). Indeed, these tumors showed increased angiogenesis (ERG positive), increased proliferation (Ki67 positive), and increased amounts of cytokines (CXCL1, 2 and 3) and immunomodulatory molecules (IL8). Again, LKB1$^{Y49D}$ showed nuclear localization within tumor cells (Fig. 6c). Loss of tumor suppressor function in the LKB1$^{G135R}$ and LKB1$^{D194Y}$ mutants was also reflected in vivo. Tumor growth was particularly promoted by the LKB1$^{G135R}$ mutant which also showed increased production of vimentin. According to the Ki67 proliferation marker both mutants promoted tumor cell proliferation (Fig. 6d, e). We did not observe metastasis in any of the models tested. Thus, the in vitro tumor suppressor capabilities of the investigated LKB1 mutants were reflected in vivo. Additionally, tumors harboring the LKB1$^{Y49D}$ mutation showed phenotypic features compatible with the molecular profile induced by this isoform.

## Discussion

Loss of function of the tumor suppressor *STK11* (LKB1) has been observed in many types of cancer. However, LKB1 also plays a role in a number of pathways involved in controlling metabolism, cell growth, angiogenesis, adhesion, and motility; even recently, LKB1 has been related to immunotherapy responses. Therefore, it is likely that the tumorigenic potential of LKB1 mutations is mediated through alternative mechanism/s. Here, we studied the contribution of four LKB1 somatic mutations to tumorigenesis in vitro and in vivo, confirming the role of LKB1 mutants in pro-tumorigenic processes such as adhesion, motility, angiogenesis, and inflammation.

As a tumor suppressor, LKB1 is usually lost in human cancer. Thus, selected missense LKB1 mutations should contribute with either an additional useful feature/s for the tumor or loss of the tumor suppressor function of the protein. Three out of the four LKB1 mutants studied, failed to function as tumor suppressors in three different cell lines from three different tumor types (lung cancer, melanoma, and cervical cancer). However, LKB1$^{R87K}$ behaved as the wild-type isoform in controlling cell proliferation. In fact, while the kinase activity of LKB1$^{R87K}$ was comparable to

that of the wild-type isoform, the enzymatic activity in LKB1$^{Y49D}$, LKB1$^{G135R}$, and LKB1$^{D194Y}$ was almost extinct, even in the presence of STRADα[14]. In this regard, mutations at residue Asp194 affect the conserved DLG triplet lying in the activation loop that helps to orientate the γ phosphate of ATP for transfer[34]; in fact LKB1$^{D194A}$ is considered a dead kinase. However, LKB1$^{D194Y}$ still has ~20% of the in vitro LKB1$^{WT}$ kinase activity, possibly because the deprotonated OH group of the Tyr residue still has a negative charge that might help to conserve some amount of activity. Residue Gly135 is also located in the ATP binding cleft; thus, it is tempting to speculate that substitution of a Gly for a charged amino acid might affect the ATP binding to LKB1 and, consequently its catalytic activity. Importantly, the described kinase activity of the different isoforms correlated with the metabolic stress response showed by the different cell lines expressing the different isoforms, according to the amounts of the direct LKB1 target AMPK.

The subcellular localization and activity of LKB1 is controlled through its interaction with STRADα and the armadillo repeat-containing mouse protein 25 (Mo25)[13,14]. All mutants except LKB1$^{Y49D}$ showed nuclear-cytoplasmatic localization. LKB1$^{Y49D}$ localized in the nucleus and had impaired activity, which is in agreement with its diminished binding capability to STRADα. Interestingly, recent studies support that LKB1$^{Y49D}$ mutation promotes variations in the binding energy pertaining to spatial conformation and flexibility, impairing the binding to STRADα and MO25[35]. Thus, the tumor suppressor activity linked to LKB1 kinase activity could be acquired through STRADα-dependent (LKB1$^{G135R}$ and LKB1$^{D194Y}$) or STRADα-independent mechanisms (LKB1$^{Y49D}$); the later also affect the subcellular localization and most likely other processes, such as transcriptional regulation[6]. In addition, LKB1$^{Y49D}$, LKB1$^{R87K}$, and LKB1$^{D194Y}$ showed a significantly shorter half-life than LKB1$^{WT}$, and LKB1$^{R87K}$ was the isoform showing the shortest half-life. In this case, the substitution of the Arg residue by a Lys residue could promote post-translational modifications that might affect the protein stability.

Beyond its tumor suppressor activity, it is known that LKB1 serves as a FAK repressor to stabilize focal adhesion sites, contributing to cell adhesion and motility[31]. Both LKB1$^{R87K}$ and LKB1$^{D194Y}$ failed to repress FAK phosphorylation compared to the wild-type isoform. Interestingly, the LKB1$^{R87K}$ mutant promoted cell motility together with cytoskeletal reorganization. This effect on cell motility was also induced by LKB1$^{D194Y}$, although it was less significant than the effects seen in LKB1$^{R87K}$ cells. In relation to this, it is known that LKB1 induces the degradation of β-catenin, a molecule involved in cell–cell contact in epithelial cells[36]. Both LKB1$^{R87K}$ and LKB1$^{D194Y}$ promoted more β-catenin degradation than the wild-type counterpart and induced the downregulation of *TGFB1I1* and *CDCP1*, two molecules also involved in adhesion and motility[37,38].

It is clear that reconstitution of LKB1 implies changes in the transcriptional profile of cells that were supported by previous

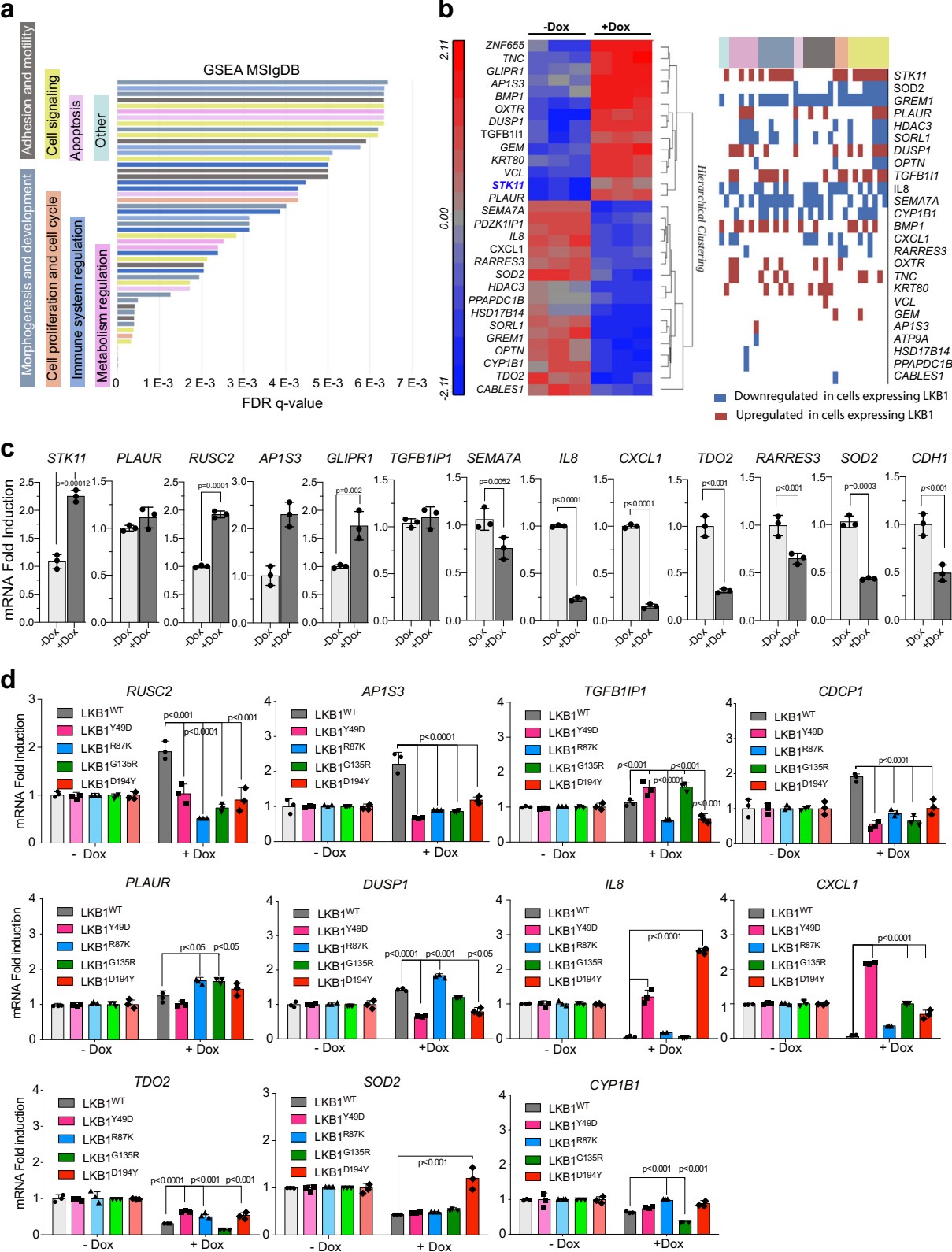

**Fig. 4 Vesicle trafficking regulating molecules, adhesion molecules, and cytokines are differentially regulated by LKB1 mutant isoform. a** Gene set enrichment analysis of genes showing >1.2-fold difference ($p < 0.05$) in A549 cells expressing LKB1 compared to non-LKB1 expressing cells. The most significant ontology terms are shown in different colors. **b** Heat map of the unsupervised hierarchical clustering of the top regulated genes (FDR < 0.1) upon LKB1 expression. On the right the classification of the top regulated genes within the ontology terms (same code color as in **a**) is shown. **c** Validation by RT-PCR of genes regulated upon LKB1$^{WT}$ expression. The graphs show the mean of three different experiments. The $p$-values were calculated by Student's $t$ test. **d** RT-PCR of the indicated genes upon the expression of the different LKB1 isoforms (+doxycycline). The graphs show the mean of three different experiments. $p$-values were calculated by Student's $t$ test.

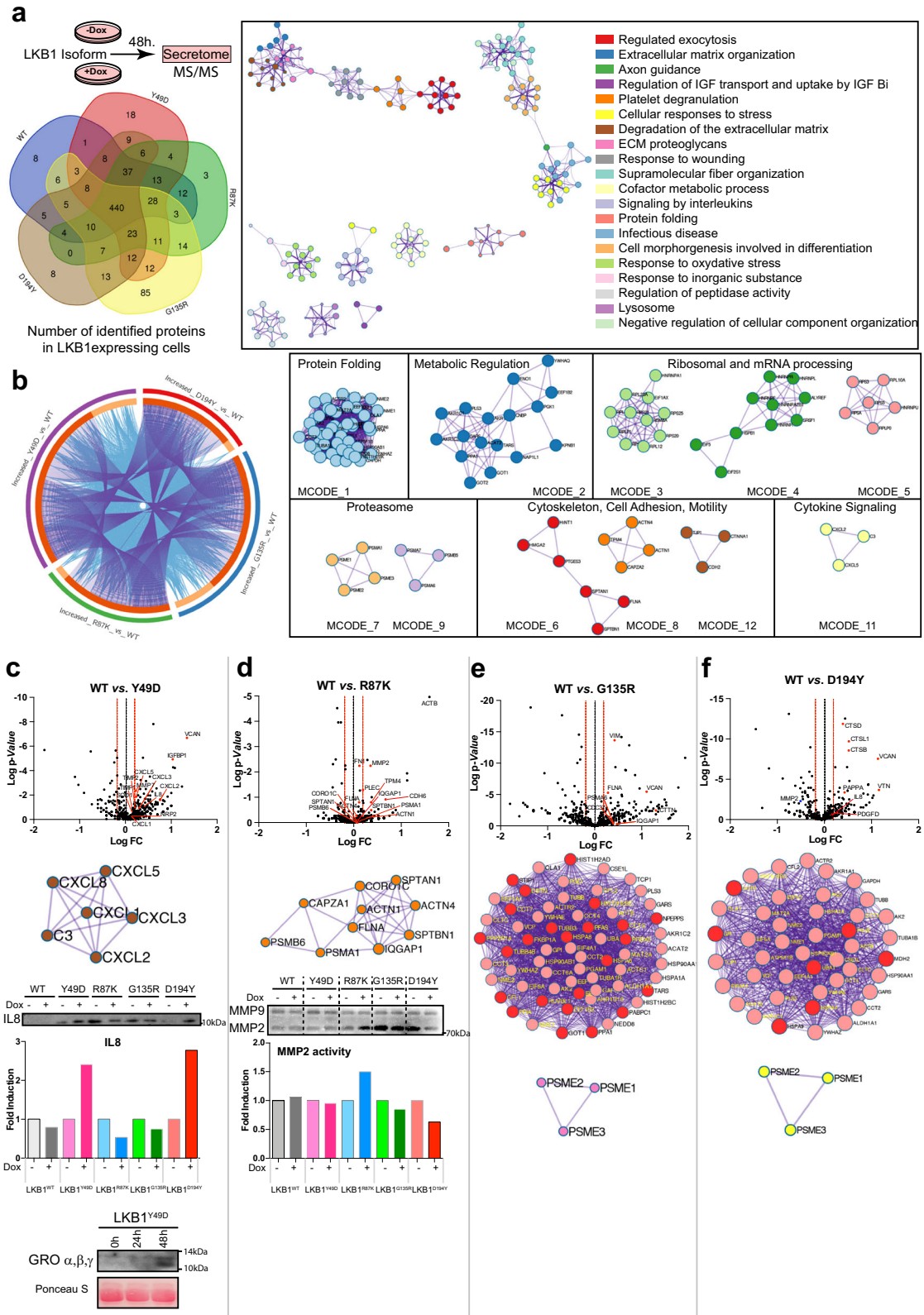

studies[33]. Some genes involved in vesicle trafficking, regulation of autophagy, and inflammation (*RUSC2* and *AP1S3*) failed to be upregulated by the mutant isoforms, which might have some effects in autophagosome maturation and distribution, as well as proinflammatory consequences[39–42]. The latter, could be more prominent in LKB1[Y49D] expressing cells that showed a stronger dysregulation of *AP1S3*[41,42]. Another particularly interesting set

of regulated genes, given to their clinical relevance, is cytokines (i.e., *IL8* and *CXCL1*) which become downregulated upon *STK11* expression. Our data suggest that loss of LKB1 and/or somatic mutation of this protein in tumor cells might lead to the upregulation of these cytokines contributing to the deregulation of the immune response and tumor angiogenesis. In particular, both LKB1[Y49D] and LKB1[D194Y] not only failed also repress *IL8* and

**Fig. 5 Secretome analysis induced by LKB1 isoforms identified differential alterations in exocytosis regulation, cell adhesion and motility, and cytokine signaling. a** The recovered conditioned media from A549 parental and LKB1-isoform induced cells (after 48 h) was analyzed by mass spectrometry. Venn diagram showing the common proteins identified in the secretomes obtained from cells expressing the different LKB1 isoforms. On the right, a functional enrichment and interactome analysis of all the protein identified after LKB1-isoform expression is shown. The intracluster and intercluster similarities of enriched terms are shown. Cluster annotations are shown in color coded. (metascape.org). **b** Circos plot showing the overlap between protein lists (increased proteins in the LKB1 mutant isoforms vs. LKB1$^{WT}$). The purple curves indicate overlap at the protein level and the blue curves link proteins that belong to the same enriched ontology term. On the right the protein–protein interaction network and the MCODE component[48] identified in the gene lists are shown. **c** Volcano plot depicting protein data p-values vs. fold change after comparison of proteomics data from LKB1$^{WT}$ vs. LKB1$^{Y49D}$ cells. The most representative protein–protein interaction network is shown. Below, western blot showing the levels of IL8 and CXCL1, CXCL2, and CXCL3 in A549 cell's conditioned media after 48 h of LKB1 induction (IL8) or at the indicated time points. The graphs show quantification of the bands. Ponceau S was used as a loading control. **d** Same as in **c**, showing data from LKB1$^{WT}$ vs. LKB1$^{R87K}$ cells. Below, zymogram showing the MMP9 and MMP2 activity in conditioned media upon induction of the different LKB1 isoforms. The graphs show quantification of the bands. **e, f** Same as in **c** showing data from LKB1$^{WT}$ vs. LKB1$^{G135R}$ and LKB1$^{WT}$ vs. LKB1$^{D194Y}$ cells, respectively. In the protein network, pink circles depict shared proteins between two sets of samples and red circles represent proteins detected only in each set.

*CXCL1* transcription but induced their transcriptional regulation. Vascular abnormalities in part due to deregulation of VEGF have been described in LKB1-deficient mice. Our observations suggest that loss of *STK11* and/or LKB1$^{Y49D}$ and LKB1$^{D194Y}$ somatic mutations in tumor cells could also contribute to angiogenesis and immune response through upregulation of cytokines. This piece of data could be especially relevant in relation to the clinical responses to immunotherapy observed in LKB1-deficient *KRAS*-mutated lung tumors[43]. TDO2 an enzyme catalyzing the production of kynurenine, which promotes immune-tolerant dendritic cells (DCs) and regulatory T cells and thus contributes to an immunosuppressive environment, is also transcriptionally upregulated through the loss of *STK11* or mutated isoforms. The contributions of these molecules to immunotherapy responses in a *STK11*-deficient or *STK11*-mutated context are currently under investigation.

Comparative secretome analysis of samples not only confirmed the role of LKB1 in regulating processes involved in motility and cell adhesion such as extracellular matrix organization and morphogenesis, but also revealed its participation in processes such as vesicular transportation and cytokine production. In fact, these results support both the possible role of LKB1$^{Y49D}$ in regulating IL8 and CXCL1, 2, 3, and 5, and promoting inflammation and angiogenesis, and the contribution of LKB1$^{R87K}$ to cell motility and adhesion through the regulation of extracellular matrix-remodeling molecules such as MMP2 and protein networks related to this phenotype. LKB1$^{G135R}$ and LKB1$^{D194Y}$ mutants showed certain similarities: both mutants lost the tumor suppressor capability of LKB1 and both mutations affect the LKB1 ATP binding cleft. In addition, the secretome analysis also identified similar protein–protein networks for these two isoforms, including processes networks involved in antigen processing, which could be relevant for immunotherapy.

Notably, the tumor suppressor activity of LKB1$^{WT}$ and LKB1$^{R87K}$ was also observed in vivo. Unfortunately, we did not observe any increased metastasis promoted by LKB1$^{R87K}$. Interestingly, all tumors expressing LKB1$^{Y49D}$ showed signs swelling supporting a role of LKB1$^{Y49D}$ in regulating cytokine production and inflammation-related processes. These data were braced by an increased expression of IL8 and GROα, β, and γ (CXCL1, 2, and 3) in these tumors compared to the expression of these in parental cells or LKB1$^{WT-}$expressing tumors. LKB1$^{G135R}$ and LKB1$^{D194Y}$ not only did not function as tumor suppressors but also the promoted tumor growth. Since parental cells lack the tumor suppressor activity of LKB1, these results suggest that LKB1$^{G135R}$ and LKB1$^{D194Y}$ mutants promote in vivo tumor growth by a mechanism independent of their kinase activity, a result that is reflected in the proliferation index of the tumors (Ki67 staining). In agreement with this, it has been described that

LKB1 mutants that are catalytically deficient enhance cyclin D1 expression[6], which contributes to tumor growth. This finding also supports the ability of these mutants to differentially regulate the expression of specific genes (i.e., LKB1$^{Y49D}$ regulation of IL8 and CXCL1) that might contribute to tumor development and progression.

In summary, we show that beyond the role of the non-mutated protein as a tumor suppressor, missense LKB1 somatic mutations could contribute to tumor development and/or progression by modifying not only intrinsic cell capabilities such as proliferation, motility, or adhesion but also the tumor microenvironment, affecting inflammatory responses and likely the immune system. Interestingly, these effects can be both kinase dependent and kinase independent, unveiling possible roles for LKB1 independent of its enzymatic activity. These results will contribute to clarify the unknown significance of missense somatic LKB1 mutations in human cancer, assisting with the diagnosis of diseases to help guide optimal treatment.

## Methods

**Reagents.** Doxycycline, cycloheximide, and Ponceau S solution were obtained from Sigma-Aldrich Quimica (Madrid, Spain). Horseradish peroxidase and secondary fluorescent antibodies were obtained from GE Healthcare (Little Calfont, UK) and Thermo Scientific (Fremont, CA, USA), respectively. FITC-phalloidin was purchased from Abcam (Cambridge, UK); antibodies against LKB1 (ley37D/G6), IL-8, and GROα,β,γ (CXCL1, CXCL2, and CXCL3) were obtained from Santa Cruz Biotechnology (Heidelberg, Germany); p-AMPKα, AMPKα, Anti-LKB1 (D60C5F10), and Anti-β-catenin antibodies were obtained from Cell Signaling (Leiden, The Netherlands); anti-Ki67 antibodies were purchased from Abcam (Cambridge, UK); and anti-vimentin and anti-ERG antibodies were obtained from ROCHE, Ventana (Basel, Switzerland). Anti-E-cadherin antibodies were purchased from R&D Systems (Minneapolis, MN, USA). Anti-GAPDH antibodies were purchased from Trevigen (Gaithersburg, MD, USA).

**Construct generation.** The pLenti-rtTA2-IRES-H2B-GFP doxycycline-inducible plasmid was obtained from S. Tenbaum, HG Palmer's Lab (Vall d´Hebron Institute of Oncology, VHIO). The human LKB1 sequence was subcloned from pCMV5-Flag-LKB1$^{WT}$ to obtain pLenti-rtTA2-LKB1$^{WT}$-IRES-GFP. The different mutant isoforms were generated via site-directed mutagenesis using the Quick-Change II Kit (Stratagene, Cedar Creek, TX, USA) and the following primers: LKB1$^{Y49D}$: ATCGGCAAG**GAC**CTGATGGGG, LKB1$^{R87K}$: AAGTTGCGA**AAG**ATCCCCA AC, LKB1$^{G135R}$: GCGTGTGT**CGC**ATGCAGGAAA and LKB1$^{D194Y}$: AAAATCT CC**TAC**CTGGGCGTG.

**Cell culture.** A549 lung cancer cells, G361 melanoma cells and HeLa cells, all of which were null for *STK11* mutations, were obtained from ATCC. Cells were grown in Dulbecco's modified Eagle medium (DMEM) (Biowest, Riverside, MO, USA) supplemented with 10% fetal bovine serum (FBS) (Biowest) and 100 μg/mL penicillin/streptomycin (Thermo Scientific, Waltham, MA, USA) and maintained at 37 °C and 5% $CO_2$. Cells were infected with the doxycycline-inducible construct rtTA2-H2B-GFP containing the different isoforms of LKB1 (*LKB1*$^{WT}$, *LKB1*$^{Y49D}$, *LKB1*$^{R87K}$, *LKB1*$^{G135R}$, or *LKB1*$^{D194Y}$). Cells were induced with 1 μg/mL doxycycline (Sigma) for 48 h, and green fluorescent cells were sorted via a FacsAria Digital Cell Sorter (BD Biosciences, San Jose, CA, USA). For glucose starvation

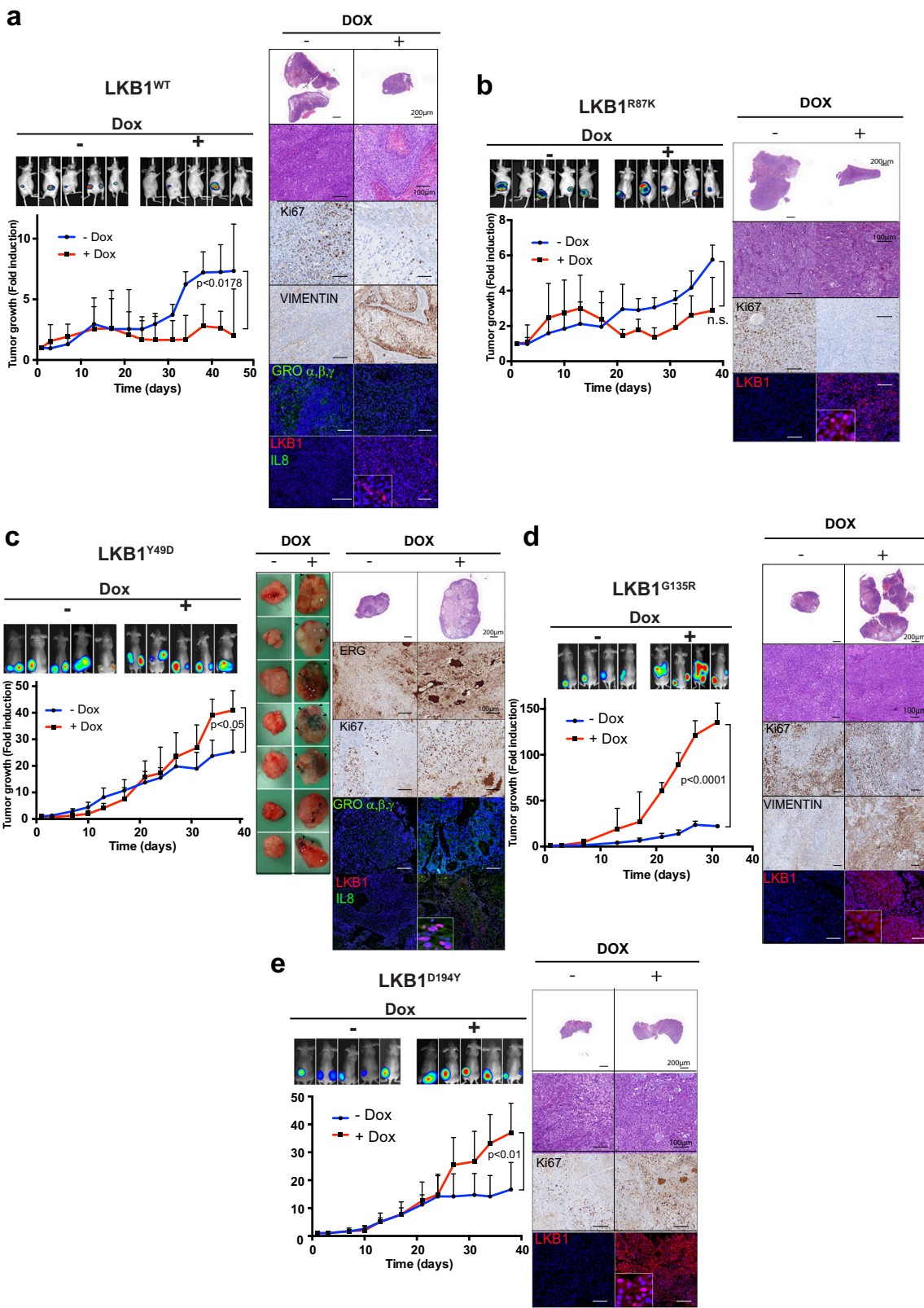

experiments cells were culture in DMEM without glucose for 3 h before total protein was recollected.

**Immunoblots.** Cells were lysed in RIPA lysis buffer, equal amounts of protein were subjected to SDS-PAGE and transferred to a PVDF membrane. Immunoblots were performed as previously described[44,45].

**Quantitative-reverse transcriptase polymerase chain reaction (qRT-PCR).** Two hundred micrograms of RNA per sample was used to generate cDNA using the SuperScript III First-Strand Synthesis System for RT-PCR Kit (Invitrogen, Carlsbad, CA, USA). Quantitative PCR analysis was performed using the SYBR Green PCR Master Mix Kit (Applied Biosystems Inc., Foster City, CA, USA) and the ABI Prism 7900HT Fast Real-Time PCR System (Applied Biosystems Inc.). The primers used are shown in Table 1. The measurements were calculated by employing the ΔΔCt method using SDS 2.3 Software (Applied Biosystems, Inc.).

**Fig. 6 Distinct contributions of LKB1 mutant isoforms to in vivo tumor growth. a** In vivo tumor growth upon expression of LKB1$^{WT}$ ($n = 5$ per group). IVIS imaging of tumor growth at 45 days is shown. H&E staining of representative tumors, immunohistochemistry images from Ki67 staining and vimentin staining, and immunofluorescence images of LKB1, IL8, and GROα,β,γ (CXCL1, CXCL2, and CXCL3) are shown. **b** In vivo tumor growth upon expression of LKB1$^{R87K}$ ($n = 5$ per group). The representative images of H&E staining, immunostaining for Ki67 and immunofluorescence staining for LKB1 are shown **c**. In vivo tumor growth upon expression of the LKB1$^{Y49D}$ mutant ($n = 5$ per group). Pictures of fresh tumors are shown. Arrows point to hemorrhagic areas. The representative images of immunostaining and immunofluorescence for ERG and Ki67, and LKB1, IL8 and GROα,β,γ (CXCL1, CXCL2, and CXCL3), respectively, are shown. **d** In vivo tumor growth upon expression of LKB1$^{G135R}$ ($n = 4$ per group). The representative images of H&E staining, immunostaining for Ki67 and vimentin, and immunofluorescence staining for LKB1 are shown. **e** In vivo tumor growth upon expression of LKB1$^{D194Y}$ ($n = 5$ per group). The representative images of H&E staining, immunostaining for Ki67 and immunofluorescence staining for LKB1 are shown. The *p*-value was calculated by a two-tailed binomial test. Bars represent 500 μm.

### Table 1 List of primers used for qPCR.

| Gene | FWR/REV | Sequence-5′-3′ |
| --- | --- | --- |
| STK11 | FWR | TCTACACTCAGGACTTCACG |
|  | REV | GTTCATACACACGGCCTT |
| PLAUR | FWR | CATGAATCAATGTCTGGTAGCC |
|  | REV | GCCTCTTACCATATAGCTTTG |
| RUSC2 | FWR | ATTTCCATTGACCTGCTTCAG |
|  | REV | CTTGTGCCAAAATGAGCC |
| AP1S3 | FWR | CTGTGCAATAGAAAATCAGGAC |
|  | REV | AGCTCACAGACATTTCCA |
| GLIPR1 | FWR | GCCCCAATAATGACAAGTG |
|  | REV | TTTGACTTGGTCTCGCTG |
| TGFB1I1 | FWR | GCCACTCAGTTCAACATCAC |
|  | REV | TCCTCCTTCTGCTCTCCT |
| SEMA7A | FWR | TCTCAATGTGTCCCGTGT |
|  | REV | TGAACTTTCCCCACCCTG |
| IL8 | FWR | CTGAGAGTGATTGAGAGTGGAC |
|  | REV | TTTTCCTTGGGGTCCAGA |
| CXCL1 | FWR | TCAATCCTGCATCCCCCATA |
|  | REV | TTCCTCCTCCCTTCTGGT |
| TDO2 | FWR | ACTTCTGGGGAAAGCTTG |
|  | REV | GTTCCTCTTTTTCTTCAGACTC |
| RARRES3 | FWR | TGAGCACTTTGTCACCCA |
|  | REV | CACACCGACTTCAACCTT |
| SOD2 | FWR | GACAAACCTCAGCCCTAA |
|  | REV | CAGCTTCTCCTTAAACTTGTC |
| CDH1 | FWR | AGAACGCATTGCCACATACACTC |
|  | REV | CATTCTGATCGGTTACCGTGATC |
| CDCP1 | FWR | GACGGTGTCCTTTATACC |
|  | REV | GACCTTGCTTTTTGTGTCAG |
| DUSP1 | FWR | AGGACAACCACAAGGCAG |
|  | REV | TGGACAAACACCCTTCCT |
| PLAUR | FWR | CATGAATCAATGTCTGGTAGCC |
|  | REV | GCCTCTTACCATATAGCTTTG |
| CYP1B1 | FWR | TGCTCCTCCTCTTCACCA |
|  | REV | GGTCACCCATACAAGGCA |

We applied geNorm algorithms to select TATA-binding protein (TBP) and human peptidyl-prolyl cis-trans isomerase A (HPPIA). to select TATA-binding protein (TBP) and peptidylprolyl isomerase A (cyclophilin A, PPIA) as the most stable reference transcripts. The geometric means of the expression values for both housekeeping genes were used to normalize the expression and to calculate the normalized SD of all transcripts analyzed. Relative expression levels were calculated after normalization. Data were represented as mean ± SD of triplicates from three independent experiments (biological replicates).

**Gene expression analysis.** For microarray analyses, we used a genome-wide Human Gene 1.0 ST Array (Affymetrix, Santa Clara, CA, USA). Genes were considered differentially expressed in A549-LKB1-WT cells if the fold change was >1.2 and the *p* was <0.05 (noninduced cells versus cells treated with doxycycline for 48 h) using a two-tailed one-way ANOVA test.

**Proliferation and colony-formation assays.** We seeded $0.16 \times 10^6$ cells per well in 6-well plates. The data were collected in triplicate at 24, 48, 72, and 96 h after the initial seeding. For every time point, viable cells were counted (using a Neubauer chamber). Data are presented as the fold change with respect to the first measure

(0 h.). For colony-formation assays, 300 cells per well were seeded in triplicate for every condition tested. Cells were maintained at 37 °C in 5% CO$_2$ for ~20 days. Then, the cells were washed twice with PBS and fixed for 10 min with 4% paraformaldehyde in PBS at RT. Cells were dyed for 10 min with a crystal violet staining solution. After distaining, colonies were photographed and counted. Colony quantification was performed manually and by using ImageJ software. At least two biological replicates with three technical replicates each were performed for every cell line.

**Cell cycle analysis.** Cells were grown in complete media and treated for 48 h with doxycycline 1 μg/mL to determine LKB1-isoform expression. Time point treatments were performed in triplicate. Then, the medium and cells were collected, and after centrifugation, the cells were fixed and stained with the Cell Cycle Analysis Guava reagent (Guava Technologies, Hayward, CA, USA). Samples were analyzed with the Guava PCA cytometer (Guava Technologies Hayward, CA, USA).

**Metabolic profiling.** Mitochondrial function and glycolytic function were assessed using Seahorse technology (Seahorse XF Cell Mito stress kit, Agilent Technologies, Wilmington, DE, USA). Briefly A459 cells harboring the different *STK11* isoform constructs were cultured on Seahorse XF-24 plates at a density of 75,000 cells per well. Cells were grown in the presence of doxycycline for 4 days before experiment. On the day of metabolic flux analysis, cells were changed to unbuffered DMEM (DMEM base medium supplemented with 10 mM glucose, 1 mM sodium pyruvate, 2 mM Glutamine, pH 7.4) and incubated at 37 °C in a non-CO2 incubator for 1 h. All medium and injection reagents were adjusted to pH 7.4 on the day of assay. Four baseline measurements of OCR and ECAR were taken before sequential injection of mitochondrial inhibitors. Three readings were taken after each addition of mitochondrial inhibitor before injection of the subsequent inhibitors. The mitochondrial inhibitors used were oligomycin (1 μM), FCCP (0.5 μM), and rotenone (0.5 μM). OCR and ECAR were automatically calculated and recorded by the Seahorse XF-24 software. After the assays, plates were saved and protein readings were measured for each well to confirm equal cell numbers per well. The percentage of change compared with the basal rates was calculated as the value of change divided by the average value of baseline readings.

**Motility assay.** Cells were seeded at low confluence in 24-well plates in duplicate. After 24 h, cells were induced with doxycycline and placed in the IncuCyte Live Cell Analysis Platform (Essen, Ann Arbor, MI, USA). Pictures were captured at 30 min intervals from five separate 950 × 760 μm$^2$ regions per well using a ×20 objective for 5 days. Motility rates were measured for 30 to 50 GFP-positive cells individually with ImageJ software and graphed.

**Kinase assay.** A specific LKB1 kinase assay was performed as previously described in Lizcano et al.[15]. Briefly, different combinations of His-tagged LKB1 isoforms and FLAG-tagged STRADα were expressed in 293 cells and the complexes purified on cobalt binding resin. Protein complexes were washed twice with 1 ml of lysis buffer (50 mM Tris/HCl pH 7.5, 1 mM EGTA, 1 mM EDTA, 1% (w/v), Triton-X 100, 1 mM sodium orthovanadate, 50 mM sodium fluoride, 5 mM sodium pyrophosphate, 0.27 M sucrose, 0.1% (v/v) 2-mercaptoethanol and 'complete' proteinase inhibitor cocktail (one tablet/50 ml) containing 0.5 M NaCl, and twice with 1 ml of Buffer A (50 mM Tris/HCl pH 7.5, 0.1 mM EGTA, and 0.1% (v/v) 2-mercaptoethanol) and eluted from column with elution buffer containing 50 mM sodium phosphate, 300 mM NaCl and 150 mM imidazole. Phosphotransferase activity towards the NUAKtide peptide (SNLYHQGKFLQTFCGSPLYRRR residues 241–260 of human NUAK2 with three additional Arg residues added to the C-terminal to enable binding to P81 paper), was then measured in a total assay volume of 50 μl consisting of 50 mM Tris/HCl pH 7.5, 0.1 mM

EGTA, 0.1% (by vol) 2-mercaptoethanol, 10 mM magnesium acetate, 0.1 mM [γ32P]ATP (~200 cpm/pmol), and 200 μM NUAKtide peptide. The assays were carried out at 30 °C with continuous shaking, to keep the immunoprecipitates in suspension, and were terminated after 10 min by applying 40 μl of the reaction mixture onto p81 membranes. The p81 membranes were washed in phosphoric acid, and the incorporated radioactivity was measured by scintillation counting.

**Secretome proteomics and statistical analysis.** The secretomes were prepared as previously described[46]. In brief, $4 \times 10^6$ cells were seeded in 150 cc tissue culture plates and allowed to grow for 48 in the presence or absence of doxycycline 1 µg/ml. After that, media was aspirated and cells were washed twice with PBS and then three times with serum-free media. Then, cells at 60–70% confluency were maintained in serum-free media for 24 h before the collection of the conditioned media (secretome). Secretomes were spun down, and filtered through a 0.22-µm pore filter. Then, secretomes were concentrated using a 10,000 MWCO Millipore Amicon Ultra filter (Millipore) until a final volume of 50 µL was reached. The protein concentration was determined using the Pierce BCA Protein Assay kit (Thermo Scientific). All samples were digested with trypsin in-solution prior to analysis by liquid chromatography−mass spectrometry (LC − MS) as previously described (1). Tryptic digests were analyzed by shotgun proteomics using an LTQ Velos-Orbitrap mass spectrometer (Thermo Fisher Scientific, Bremen, Germany). The RAW files of each MS run were processed using Proteome Discoverer (Thermo Fisher Scientific), and MS/MS spectra were searched against the human database of Swiss-Prot using the MASCOT (Matrix Science, London, U.K) algorithm. The results files generated from MASCOT (.DAT files) were then loaded into Scaffold (Proteome Software, Portland, OR), resulting in a nonredundant list of identified proteins per sample achieving a protein false discovery rate (FDR) under 1.0%, as estimated by a search against a decoy database.

Relative spectral counting-based protein quantification analysis was performed on the different samples analyzed using Scaffold. Files containing all spectral counts for each sample and its replicates were generated and then exported to R software for normalization and statistical analysis[47]. All statistical computations were done using the open-source statistical package R. The data were assembled in a matrix of spectral counts, where columns represent the biological conditions and rows represent the identified proteins. An unsupervised exploratory data analysis (EDA) by means of principal components analysis and hierarchical clustering of the samples on the SpC matrix was first performed. Then, the GLM model based on the Poisson distribution was used as a significance test[47]. Finally, the Benjamini-Hochberg multiple test correction was used to adjust the p-values with control on the false discovery rate (FDR).

**Animal study.** Animal experiments were conducted and designed according to protocols approved by the Institutional Animal Care and Use Committee of Vall d´Hebron Institute of Research. Nude mice (athymic nu/nu, female 4-6weeks old, Harlan Laboratories) were used for xenograft studies.

**In vivo tumor growth experiments.** For xenograft animal models, $5 \times 10^5$ luciferase-expressing A549-LKB1 cells were subcutaneously implanted into 8-week-old female athymic nude-Foxn1[nu] mice ($n = 5$ per group) (Envigo, Indianapolis, IN, USA). Tumors were measured with a digital Vernier caliper, and the mice were weighed twice a week. Tumor volume was calculated as $D \times d2/2$, where $D$ was the major diameter and $d$ was the minor diameter. Bioluminescence imaging (IVIS Spectrum In Vivo Imaging System), PerkinElmer Life Science, Waltham, MA, USA) was performed at the end of the experiment for metastasis screening, both in vivo and ex vivo (viscera). Once the treatment was started, the animal weights and tumor sizes were monitored every 2 days. Representative tumor pictures were taken and tumor samples were obtained for further analysis.

**Immunohistochemistry and immunofluorescence.** Formalin-fixed paraffin-embedded tumor samples were subjected to immunocytochemistry according to the manufacturer's antibody protocol. Samples were developed either by using either secondary antibodies linked to horseradish peroxidase or secondary antibodies linked to a fluorophore. Immunostaining was performed on 4 µm sections from formalin-fixed paraffin-embedded tissues. Staining was performed either manually or on the automated immunostainer Beckmarck XT (Ventana Medical Systems, Roche, Tucson, AZ, USA). Antibodies were visualized by the Ultra-View[TM] Universal DAB Detection Kit (Ventana Medical Systems). For samples processed manualy antigen retrieval was performed using target retrieval solution pH 6.0 (Dako,Agilent, Santa Clara CA, USA)) Samples were scanned (panoramic slide digital scanner) and evaluated by two independent pathologists (using 3DHistech software).

**Statistics and reproducibility.** Statistical tests used are reported in the figure legends. In summary, statistical analyses were performed in GraphPad Prism 6 (GraphPad Software Inc.) using a two-tailed Student's t test to compare differences between two groups or one-way ANOVA with multiple-comparisons tests to compare 3 or more groups. Statistical tests used for the analyses of transcriptomes (microarrays and gene set enrichment analysis) were performed in Partek Genomic Suite software (Partek Inc.). Number of biological replicates was ≥3 and is indicated in the figure legends.

**Reporting summary.** Further information on research design is available in the Nature Research Reporting Summary linked to this article.

## Data availability

The datasets generated during and/or analyzed during the current study are available in: ArrayExpress public repository with accession number E-MTAB-8863 (https://www.ebi.ac.uk/arrayexpress/experiments/E-MTAB-8863/) and the proteomeXchange public database PRIDE (https://www.ebi.ac.uk/pride/archive) with accession number: PXD018041.

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

## Acknowledgements

This work was supported by funds from the Spanish Health Ministry (Fondo de Investigaciones Sanitarias-FIS) PI1400375-Fondos FEDER, PI17/00043-Fondos FEDER, Euronanomed2-ISCIII (AC16/00019)-Fondos FEDER, Asociación Española Contra el Cancer (AECC-GCB15152978SOEN) supported PGM, KM., Ramón Areces Foundation supported KM and research.

## Author contributions

Conceptualization: J.A.R., P.G.M., and E.G.S. Investigation: P.G.M., E.G.S., S.G.O., K.M., N.N.G., A.C.R., R.S., J.H.L., B.F., J.V., O.M., S.E.G., J.M.L. and J.A.R. Resources: E.M.C. and V.G.P. Methodology: J.G. and R.G. Formal analysis: J.A.R., J.M.L., F.C., J.V. Writing-review and editing: J.A.R., P.G.M., S.G.O., and K.M. Supervision: J.A.R.

## Competing interests

The authors declare no competing interests.
