## [Peer Review File · Communications Biology]

Reviewers' comments:

Reviewer #1 (Remarks to the Author):

This is a timely and well-written report on the functional characterization of four STK11 mutations in tumor cells. The manuscript is interesting because STK11(LKB1) is often somatically mutated (or deleted) in NSCLC and its alterations have been associated with poor response to immunotherapy with ICI (as well as to metabolic alterations). Since LKB1 mutations are not concentrated in hotspots, there are many cancer-associated mutations which remain poorly investigated in terms of their impact on tumor cell functions.

A few points which deserve to be improved:

1. Can the Authors explain in the Introduction how they selected the 4 missense LKB1 somatic mutations investigated in this study?
2. With regard to the study design and the methodology used here, one concern is that by their lentiviral vector-mediated approach the Authors could actually over-express LKB1 mutants, compared with levels found in tumor cells bearing these mutations. Some biological effects measured here could be "dosage dependent". Can the Authors provide some evidence of the relative expression levels of LKB1 in their tumor cells compared with "physiological" levels detected in LKB1-mutant tumor cells?
3. Page 13, chapter "Vesicle trafficking regulatory molecules.." It is not clear which tumor cells are used in this set of experiments. Please correct the text accordingly.
4. With regard to the translational relevance of these findings, is any of these 4 mutations being reported in patients treated with ICI? Do they associate with resistance to ICI?
5. LKB1 is best known for its effects on several metabolic pathways. It is surprising to see that the Authors did not perform any metabolic assay with these LKB1 variants. At a minimum, Seahorse analysis should be performed and glycolysis/OXPHOS activity in the cell lines expressing the mutants compared with LKB1 WT or null cells should be shown. Also, since LKB1 mutations have been associated with response to Metformin in vitro, could the Authors evaluate this in their experimental system? In my opinion, these results would complete their comprehensive functional characterization of these mutants.

Reviewer #2 (Remarks to the Author):

In the manuscript entitled "STK11 (LKB1) missense somatic mutant isoforms promote tumor growth, motility and inflammation" authors explore the biological implication of four somatic mutations for the tumor suppressor gene STK11. Authors found that mutation in Y49D, G135R and D194Y increases proliferation, tumor growth and reduce kinase activity, while R87K mutation displays a similar phenotype to wt isoform. This greater tumorigenic phenotype is explained, only in part, by higher motility and modulation of vesicle trafficking, adhesion regulation or cytokines production. STK11, is the third most commonly altered gene in lung adenocarcinoma, and it is also a risk factor in other type of tumors such as pancreas, gastrointestinal, breast, cervical, uterine, and testicular cancer. The STK11 gene encoding LKB1 protein is emerging as an important tumor suppressor that may impact the metastatic propensity of tumors as well as therapeutic response. Most mutations are truncating or loss of function so understanding the functional impact of different mutations is a relevant issue. The study has some interesting data suggesting that there may be different mechanisms by which LKB1 function is lost including increased degradation vs reduced function of the kinase.

The manuscript has some significant shortcomings that limit the overall impact of the study, however. Significant issues include the following:

5. The clinical significance of these particular mutations is unclear. In searching through the Lung adenocarcinoma TCGA, I did not see any examples of R87K, Y49D, or G135R and just a few cases of D194Y. In looking at common inherited mutations in Peutz Jager syndrome I did not see these mutations occur commonly (e.g. Amos et al, J Med Genetics 2004). The authors should clarify: What is the rationale for these particular mutations? What percentage of lung cancer or other cancer patients are affected by these particular mutations? This data should be included in the manuscript of supplement.

6. Comparison of gene expression changes (e.g. Fig 4, Supp Fig S3) seems to be done comparing triplicate copies of a single cell line expressing the particular mutant. This makes it difficult to know to what extent the changes in gene expression are truly related to the particular mutation or an artefact specific to the cell line expressing a particular mutant at a particular expression level (also note that A549 have other mutations such as KEAP1 and may behave differently at times than other LKB1-deficient cells). Given that there are publicly available datasets and robust gene expression signatures for LKB1-deficient tumors (e.g. Kaufman et al JTO 2014), it would be more appropriate to A) focus on genes known to be regulated by LKB1 in clinical datasets, and then B) Assess whether those particular LKB1-regulated genes are differentially regulated by different mutants, ideally using clinical data or at least more than one model expressing a particular mutation. There are similar concerns regarding the comparisons of the secreted proteins.

7. The significance of the growth rates in the in vivo models is unclear because we see that the growth rates of the control group (no doxycycline and hence no mutant expression) should be similar across the pairs (since it should be an identical LKB1 null background); in fact there is wide variability with about 6 fold growth in the R87K experiment (Fig 6C) and more than 20 fold growth in the Y49k experiment. This suggests that there could be some leakiness in the dox promoter or that there are simply differences in growth rates across the different stable cell lines. Therefore basing comparisons in the growth with or without dox when the baseline is different for each does not seem reasonable. It would be more appropriate to see if similar mutation-specific differences are observed across different models expressing the particular mutant.

8. LKB1 is a master kinase that activates several downstream pathways by phosphorylating different substrates. In term of characterization of loss of function and kinase activity in the different mutant isoforms, data provided in the manuscript should be further supported by assessing LKB1 downstream targets such as AMPK/mTOR (cell proliferation and metabolism control) and/or MARK/SAD/SIK (cell polarization) should be performed. Recent papers (Hollstein et al, Cancer Discovery 2019 and Murray et al Cancer Discovery 2019) are highly relevant in this regard and should be cited.

9. In vitro assays in figure 1, as well as, in vivo experiments shown in figure 6 revealed a potentially interesting result: expression of the G135R and D194Y mutations in an LKB1 deficient cell line (A549), accelerated tumor growth compared with parental cells, which already has a high proliferative phenotype due to lack of LKB1 expression. This would suggest that the mutations could activate pathways beyond simply LKB1 loss. This should be evaluated in a different model and the authors should discuss possible explanations for this phenomenon.

The finding that some mutants may accelerate tumor growth (e.g. Figure 6D) compared with the null background does not seem to fit with the earlier in vitro experiments showing that the mutants did not suppress tumor growth but did not accelerate it either (figure 1), which would be more consistent with the role of a tumor suppressor. The statement on page 16 "Thus, the in vitro tumor suppressor capabilities of the investigated LKB1 mutants were reflected in vivo." is not accurate; there is no significant suppression for the R87K and there is significant acceleration in Fig 6B, D, and E. The authors should discuss the differences in these findings.

10. In figure 1, data revealed that Q135R and D194Y mutations increased cell proliferation when expressed in a LKB1 deficient cell line (A549), while R87K and wt isoforms reduced cell growth. By contrast, Y49D mutant isoform seems not to have effects in term of cells proliferation. In figure 2, by

contrast, Q135R and D194Y only partially lost kinase activity, while Y49D also significantly reduced its kinase activity and this isoform seems to lose ability to interact with STRAD α . On the other hand, R87K (wt-like isoform), Y49D and D194K, but not G135R display a shorter protein half-life. These data are inconsistent with the proliferative phenotype shown in figure 1. The authors should discuss possible explanations for these findings.

Minor issues:

1. The statement (page 20) that "LKB1 Y49D showed signs of inflammation and disorganized angiogenesis (hemorrhagic) confirming the role of LKB1 in regulating cytokine production and inflammation" is an overstatement because the models were tested in immunocompromised mice so inflammation could not be accurately assessed; furthermore, hemorrhage is not an established marker for inflammation.
2. In Fig 1E: panel with D194Y at 48h + Dox has dark blue bar (G1 arrest) in the middle of the other two blue bars, whereas elsewhere the dark blue bar is at the bottom. Should the bar be dark blue (meaning G1 arrest), in which case it should be at the bottom, or medium blue? Also, elsewhere in the figure, there are 4 different shades of blue but only 3 in the legend- one color is nearly black. Please clarify the colors and labels.
3. The concluding statement in the Discussion is not adequately supported by the data, particularly the effects on immune modulation given the immunocompromised mouse models: "In summary, we show that beyond the role of the non-mutated protein as a tumor suppressor, missense LKB1 somatic mutations could contribute to tumor development and/or progression by modifying not only intrinsic cell capabilities such as proliferation, motility or adhesion but also the tumor microenvironment, affecting inflammatory responses and likely the immune system. "These experiments could be repeated in syngeneic murine models to better get at the impact on the microenvironment. In addition, public data could be mined (e.g. TCGA using CIBERSORT) to see if it supports the association of different patterns associated with different mutations.
4. Analysis of cells cycle phase distribution is shown in Figure 1F to support higher or lower rate of proliferation across the different mutants. Some issue with these data:
 - First, quantification of all cell cycle phases together should be close to 100%. Are the authors missing any population? For example, is SubG1 peak (Dead cells which display lower probe staining) quantified? Please, reanalyzed carefully these data.
 - Cells cycle analysis are typically performed when cells are growing exponentially to avoid include additional effects that could interfere in the cell cycle progression. Could the authors explain why G1 phase percentage is higher in -Dox treatment at 48h compared with 0h for WT and Y49D isoform? If cell cycle analysis was performed when cells are growing exponentially, G1 phase percentage at 48h should be similar to 0h. This data seems to indicate a G1 arrest at 48h due to low nutrient availability or confluent state rather than expression of LKB1 wt. Initial number of plated cells should be recalculated to allow exponential grow still at 48h.
 - Finally, averages for 2 or 3 independent experiments should be included.
5. In figure 1D and Figure S1 C authors show the quantification of clonogenic assay performed in the different mutated isoform. Are these differences statistically difference? If it does, please include this data.
6. Figure 6C is cited in the text before than Figure 6B.
7. In Discussion section, second paragraph the sentence "Thus, selected missense LKB1 selected mutations..." should be corrected by deleting the second "selected".

Reviewer #3 (Remarks to the Author):

This manuscript by Paula Granado-Martinez et al. focuses on the functional study of STK11 (LKB1) missense somatic mutant isoforms such as LKB1Y49D, LKB1R87K, LKB1G135R and LKB1D194Y in cancer. They performed several experiments including omics analysis such as secretome to reveal the functions of these mutant isoforms. This is interesting work to reveal the importance of the STK11 (LKB1) missense somatic mutant isoforms. Before it can be published in Communications Biology, I have several suggestions as described below.

1. The statistics analysis should be performed in Figure 1D, 3A and 3E.
2. The supplementary tables for proteome should be described the protein full names and the detail information about mass identification. Since many errors happened in protein identification even using software such Proteome Discoverer, to check the mass peaks carefully and list the results are important in proteome field.
3. I strongly suggest that the authors should deposit the proteome data to public database such as ProteomeXchange (<http://www.proteomexchange.org/>).

Reviewers' comments:

Reviewer #1 (Remarks to the Author):

This is a timely and well-written report on the functional characterization of four STK11 mutations in tumor cells. The manuscript is interesting because STK11(LKB1) is often somatically mutated (or deleted) in NSCLC and its alterations have been associated with poor response to immunotherapy with ICI (as well as to metabolic alterations). Since LKB1 mutations are not concentrated in hotspots, there are many cancer-associated mutations which remain poorly investigated in terms of their impact on tumor cell functions. A few points which deserve to be improved:

1. Can the Authors explain in the Introduction how they selected the 4 missense LKB1 somatic mutations investigated in this study?

We are sorry for not be more explicit in the Introduction regarding the selection of the four LKB1 mutants. First, we wanted to clarify that tumor suppressor STK11 (LKB1) is mostly lost in cancer. STK11 missense mutations represent 30% of all genetic alterations. Fifty percent of this 30% of mutations described in lung, melanoma and cervix listed in TCGA and COSMIC have been described just in one patient, another 25% in two patients and the other 25% in three or more patients. Thus, because of the numbers of patients identified so far is limited (even for the more frequent mutated residue D194G/H/N/V/Y, 42 samples (combined: lung, melanoma and cervix) / 7157 non-redundant samples-TCGA & COSMIC), it is difficult to assess the clinical relevance of these mutations. Historically, we have described several LKB1 functions from different point of views (Esteve-Puig et al., 2009; Gonzalez-Sanchez et al 2013; Esteve-Puig et al., 2014). Due to the multitask functions of LKB1, we chose these four missense LKB1 somatic mutations because we are interested in studying different functional parts of the protein contextualized in human cancer. The great majority of missense mutations in STK11, except for D194 residue, do not represent hotspots. D194N mutant was described to be kinase dead. However, the functional consequences in cancer of this mutant were mostly unknown. This residue is located in the ATP binding cleft. G135 residue is also located in the same functional region. Thus, G135R mutant was selected to be compared with D194Y since it was located in a similar 3D functional location (ATP binding cleft). Y49 residue was embedded within a β -sheet in the LKB1 N-lobe close to the 3D interaction region with STRAD. Since LKB1 subcellular localization and kinase activity is STRAD-binding dependent we were interested in study the possible consequences of this mutation. Finally, R87 is exposed at the surface of the molecule being susceptible to be modified or interact with other molecules. Moreover, the change for the lysine is particularly interesting because is not very disruptive respect the charge, however Lys and Arg have different H-bonding

capabilities and hydration free energies, and they can be post-translationally modified differently. Importantly, all selected mutations were initially detected in tumor samples and were predicted to be oncogenic or likely oncogenic (polyphen or OncoKB team TCGA). We have added a couple of sentences in the introduction explaining our motivation selecting the mutants. We hope this explanation will satisfy the reviewer concerns.

Contextualized location of selected mutations in 3D-LKB1 structure

2. With regard to the study design and the methodology used here, one concern is that by their lentiviral vector-mediated approach the Authors could actually over-express LKB1 mutants, compared with levels found in tumor cells bearing these mutations. Some biological effects measured here could be "dosage dependent". Can the Authors provide some evidence of the relative expression levels of LKB1 in their tumor cells compared with "physiological" levels detected in LKB1-mutant tumor cells?

This is a very interesting observation and at the same time is a complex and philosophical issue. We completely agree with the reviewer when state that in order to be able to interpret the results, we have to work within the "physiological range of the

protein". To sustain the expression of the protein at the "physiological range we titrated both, the concentration of the doxycycline and the time of induction in all cell lines (Supplementary figure 1). However, the physiological levels for normal cells varies according to the type of cell (melanocyte, hepatocyte...etc). In the case of tumor cells, this can be more dramatic because cells adapt to their needs or capabilities (mutations), so we find tumor cells expressing a wide range of amounts of LKB1 (See figure below showing cells expressing from no LKB1 to a variety amounts of the protein. On the right infected A459 cells uninduced and induced). In addition to this, we are talking about mutant LKB1 proteins, which adds another level of complexity. It can be the case that these mutants have their half-life altered due to different reasons (which is physiological for the selected mutant), even this altered half-life could be part of its contributions to malignancy. So, bearing all this in mind, and as showed in the figure added, we have tried to express "physiological levels" of all isoforms compared to other cancer cell lines. We have incorporated this data to the manuscript. We hope this will satisfy the reviewer concerns.

3. Page 13, chapter "Vesicle trafficking regulatory molecules." It is not clear which tumor cells are used in this set of experiments. Please correct the text accordingly.

We are sorry for not include this information. This have been corrected accordingly.

4. With regard to the translational relevance of these findings, is any of these 4 mutations being reported in patients treated with ICI? Do they associate with resistance to ICI?

As above stated missense mutations represent about 30% of the genetic mutation of STK11. Among those 50% of the punctual mutation described in lung cancer, melanoma and cervix listed in TCGA and COSMIC databases have been described in

one patient, another 25% in two patients and the other 25% have been reported in 3 or more patients. Thus, because of the numbers of patients identified so far is limited, it is difficult to assess the clinical relevance of these mutations, specially, in relation to patients subjected to immune-checkpoint inhibitors. We believe that the message in the manuscript will contribute to clarify the potential role of these mutation in patients.

5. LKB1 is best known for its effects on several metabolic pathways. It is surprising to see that the Authors did not perform any metabolic assay with these LKB1 variants. At a minim, Seahorse analysis should be performed and glycolysis/OXPHOS activity in the cell lines expressing the mutants compared with LKB1 WT or null cells should be shown. Also, since LKB1 mutations have been associated with response to Metformin in vitro, could the Authors evaluate this in their experimental system? In my opinion, these results would complete their comprehensive functional characterization of these mutants.

This is a very good question. We performed an OCR and ECAR seahorse analysis. Overall, we did not observe significant differences in OCR among the wild type isoform and the mutants. Nevertheless, the results also show that mutant G135R has significant increase in the mitochondrial proton leak that might have a major impact on mitochondrial coupling efficiency and production of reactive oxygen species. This tendency was also observed in mutants Y49D and D194Y compared to the wild-type isoform. The mechanisms of proton leak include direct movement of protons across the phospholipid membrane (the “water wires” model), diffusion through or around integral membrane proteins, or inducible transport through the adenine nucleotide translocase (ANT) or uncoupling proteins (UCP1, UCPx). Interestingly, there is a direct correlation between the metabolic rate and the proton leak. In relation to this, ECAR analysis showed a significant increase of ECAR in G135R expressing cells. This higher acidification, although not significant, was also observed in cells expressing the D194Y isoform, suggesting an increase use of glycolysis by cells expressing these two mutants. This piece of data, also correlates with the lack of tumor suppressor capabilities of these mutants and the higher proliferation rates mostly showed by G135R and D194Y expressing cell lines.

In the case of the response to Metformin, it has been shown that Metformin acts both via AMP-activated protein kinase (AMPK)-dependent and AMPK-independent mechanisms; by inhibition of mitochondrial respiration but also perhaps by inhibition of mitochondrial glycerophosphate dehydrogenase, and a mechanism involving the lysosome. Interestingly, mitochondrial dysfunction in G135R expressing cells correlated with their higher resistance to metformin.

All this information has been added to the manuscript accordingly and we hope this will satisfy the reviewer concerns.

Reviewer #2 (Remarks to the Author):

In the manuscript entitled “STK11 (LKB1) missense somatic mutant isoforms promote tumor growth, motility and inflammation” authors explore the biological implication of four somatic mutations for the tumor suppressor gene STK11. Authors found that mutation in Y49D, G135R and D194Y increases proliferation, tumor growth and reduce kinase activity, while R87K mutation displays a similar phenotype to wt isoform. This greater tumorigenic phenotype is explained, only in part, by higher motility and modulation of vesicle trafficking, adhesion regulation or cytokines production. STK11, is the third most commonly altered gene in lung adenocarcinoma, and it is also a risk factor in other type of tumors such as pancreas, gastrointestinal, breast, cervical, uterine, and testicular cancer. The STK11 gene encoding LKB1 protein is emerging as an important tumor suppressor that may impact the metastatic propensity of tumors as well as therapeutic response. Most

mutations are truncating or loss of function so understanding the functional impact of different mutations is a relevant issue. The study has some interesting data suggesting that there may be different mechanisms by which LKB1 function is lost including increased degradation vs reduced function of the kinase.

The manuscript has some significant shortcomings that limit the overall impact of the study, however. Significant issues include the following:

5. The clinical significance of these particular mutations is unclear. In searching through the Lung adenocarcinoma TCGA, I did not see any examples of R87K, Y49D, or G135R and just a few cases of D194Y. In looking at common inherited mutations in Peutz Jager syndrome I did not see these mutations occur commonly (e.g. Amos et al, J Med Genetics 2004). The authors should clarify: What is the rationale for these particular mutations? What percentage of lung cancer or other cancer patients are affected by these particular mutations? This data should be included in the manuscript of supplement.

We are sorry for not be more explicit in the Introduction regarding the selection of the four LKB1 mutants. First, we wanted to clarify that tumor suppressor STK11 (LKB1) is mostly lost in cancer. STK11 missense mutations represent 30% of all genetic alterations. Fifty percent of this 30% of mutations described in lung, melanoma and cervix listed in TCGA and COSMIC have been described just in one patient, another 25% in two patients and the other 25% in three or more patients. Thus, because of the numbers of patients identified so far is limited (even for the more frequent mutated residue D194G/H/N/V/Y, 42 samples (combined: lung, melanoma and cervix) / 7157 non-redundant samples-TCGA & COSMIC), it is difficult to assess the clinical relevance of these mutations. Historically, we have described several LKB1 functions from different point of views (Esteve-Puig et al., 2009; Gonzalez-Sanchez et al 2013; Esteve-Puig et al., 2014). Due to the multitask functions of LKB1, we chose these four missense LKB1 somatic mutations because we are interested in studying different functional parts of the protein contextualized in human cancer. The great majority of missense mutations in STK11, except for D194 residue, do not represent hotspots. D194N mutant was described to be kinase dead. However, the functional consequences in cancer of this mutant were mostly unknown. This residue is located in the ATP binding cleft. G135 residue is also located in the same functional region. Thus, G135R mutant was selected to be compared with D194Y since it was located in a similar 3D functional location (ATP binding cleft). Y49 residue was embedded within a β -sheet in the LKB1 N-lobe close to the 3D interaction region with STRAD. Since LKB1 subcellular localization and kinase activity is STRAD-binding dependent we were interested in study the possible consequences of this mutation. Finally, R87 is exposed at the surface of the molecule being susceptible to be modified or interact with other molecules. Moreover, the change for the lysine is particularly interesting because is not very disruptive respect the charge, however Lys and Arg have different H-bonding

capabilities and hydration free energies, and they can be post-translationally modified differently. Importantly, all selected mutations were initially detected in tumor samples and were predicted to be oncogenic or likely oncogenic (polyphen or OncoKB team TCGA). We have added a couple of sentences in the introduction explaining our motivation selecting the mutants. We hope this explanation will satisfy the reviewer concerns.

6. Comparison of gene expression changes (e.g. Fig 4, Supp Fig S3) seems to be done comparing triplicate copies of a single cell line expressing the particular mutant. This makes it difficult to know to what extent the changes in gene expression are truly related to the particular mutation or an artefact specific to the cell line expressing a particular mutant at a particular expression level (also note that A549 have other mutations such as KEAP1 and may behave differently at times than other LKB1-deficient cells). Given that there are publicly available datasets and robust gene expression signatures for LKB1-deficient tumors (e.g. Kaufman et al JTO 2014), it would be more appropriate to A) focus on genes known to be regulated by LKB1 in clinical datasets, and then B) Assess whether those particular LKB1-regulated genes are differentially regulated by different mutants, ideally

using clinical data or at least more than one model expressing a particular mutation. There are similar concerns regarding the comparisons of the secreted proteins.

We understand the reviewer concerns, and following his suggestion we have compared our results with all the datasets contained in Kaufman et al., JTO 2014 studying gene expression profiles associated with LKB1 loss in resected lung adenocarcinomas (7 datasets), non-small-cell lung cancer cell lines (4 studies) and murine tumors (4 mouse models). We have compared our results individually with each study, and also against all unique regulated genes from the 15 studies. Comparisons have been done using top 200 regulated genes from each study and our list of genes regulated >1.2 fold and $p < 0.05$ upon LKB1 expression (817 genes). Besides the differences between the lists in Kaufman et al., (heterogeneous tumor samples, set of patients, different groups, mouse samples... etc) when we compared individually every single dataset (200 genes) with our list (817 genes) we found between 4-15% of common regulated genes (see below figure panel A and table). These numbers were similar to the ones obtained when we compare the different human studies among them (Kaufman et al., 2014; they shared between 2-25% similarities). The number of common genes equally regulated was more frequent in the cell lines studies but also were found in the lists from the mouse models. When we compared all the unique genes regulated (after pooling the top 200 regulated genes from every single study (15 studies; 2080 unique regulated genes)) to our 817 regulated genes, we obtained 151 genes that were present in both lists including all the genes we have studied in the original version of the manuscript (see below figure panel B and table). Not only that, but some of the common genes were also associated to the 4 functional clusters described in Kaufman et al., 2014. Finally, when we asked how many of the top regulated genes from our study were common to the list of unique regulated genes from the human studies (1065 genes), we found 52 genes including genes studied in the original version of the manuscript such as (GEM, TDO2, SOD2 or ID1). We have added the regulation of three more genes in all isoforms (RT-PCR) common to the other datasets. Moreover, some of the proteins differentially regulated between the LKB1 mutants and the wild-type isoforms such as: MMP2, CXCL2, CXCL5 or TIMP1 among others, were also among the top regulated genes in Kaufman et al., 2014 study. Basically, the comparison confirmed the relevance of the genes selected in the original version of the manuscript. We hope this will satisfy the reviewer concerns. This information has been included in the manuscript accordingly.

A

B

Study #1	Study #2	Study #3	Study #4	Study #5	Study #6	Study #7	Study #8	Study #9	Study #10	Study #11	Study #12	Study #13	Study #14	Study #15
MSKCC	UNC	Wash U	Michigan	TCGA	MSKCC2	USC	Sanger	CCLC	A549	H2122	Ji (A)	Ji (B)	Carretero	Carretero Mets
FAM125B	C12orf24	ALDH3A1	FGA	ODC1	FGA	IDH2	FGA	DIO2	FGA	STC1	GDPD3	ALDH3A1	ALDH3A1	RASAL2
MEGF9	FHIT	TESC	ODC1	FGA	SLMO2	ODC1	ODC1	PTGES	PDK4	FGA	GDPD3	GDPD3	CTGF	TIMP1
MITF	FGA	PCCB	PTS	TESC	KLK10	GNAL	TNS4	TNS4	AREG	EMR2	CTSS	MMP2	FST	SOX4
RAB3B	FSTL4	CYP4F11	NEO1	DHTKD1	HBXIP	FAM105A	DIO2	PTS	CXCL2	FSTL4	ALDH3A1	MACF1	MACF1	RBM47
SECTM1	NCR2	ALDH3B1	NOL7	TBC1D2	RND1	ATP8B1	PAPSS2	FGA	TESC	GLRX	EEA1	BCL3	GEM	PDLIM7
SLC37A1	WBSR16	KIAA0319	TESC	ID1	ODC1	RPP40	ALDH3B1	PDK4	RGS2	CHAC1	TAGLN	NEO1	LGALS9	ASS1
SNX10	HOXB5	NRG1	SOD2	FSTL4	ID1	PDGFRL	STC1	TESC	TNS4	DIO2	IRF1	B3GNT1	BCL3	INCENP
TD02		GABRRAS	HBXIP	RND1	FAM105A	ALDH3A1	EMR2	RAP1GAP	FST	CREB5	UNC50	IFNGR1	ANGPTL4	IKBKE
TP53111		RNP1	ID1	ERP29	LARGE	GPRC5C	ETS2	LQC440792	CXCL5	MAML3	MACF1	MYLIP	CDCP1	RRP15
		SFN					ETS2		TRIB3	SNTB1	MMD	MKNK2	DHR33	TAGLN
		TIMM23					PDK4	EFNA1	TRIP3	ALDH3B1	OSTF1	TIMP1	NPR3	SRM
		FAM45B					PTGES	TUFT1	C5	DEPDC6	MTERFD1	CSF2RA	RELB	CKS1B
							FSTL3	KIAA0319	AKAP12	ARRB1	NF2	TNIP1	MACF1	RASA4
							TESC	MAP3K8	ANPEP	ANXA13	DUSP1	DHR33	CDC42	CYP1B1
							HBXIP	HBXIP	ANXA13	DUSP1	LARGE	MMP2	MMP2	ASPM
							RAP1GAP	CSRP1	TSPAN13	ID1	CTGF	CTGF	TNC	
							EFNA1	EFNA1	FGA	GEM	DUSP1	DUSP1	AREG	ADAM12
							DUSP1	LGR4	DIO2	SOD2				
							SLC46A3	SGK1	SGK1	PEL12				
							HSPB8	DUSP1	EEF1A2	PPFIBP2				
								DDIT3	CD55	CD55				
								SNTB1	BCL3	OSBP2				
								PLAUR	OSBP2	CEBPD				
								CD55	RAB27B	KIAA0319				
								IL8	GLRX	TRIP3				
								CDK5	CASP4					
								PPP1R15A						

Table: Common regulated genes between our study and each different study from Kaufman et al., 2014

7. The significance of the growth rates in the in vivo models is unclear because we see that the growth rates of the control group (no doxycycline and hence no mutant expression) should be similar across the pairs (since it should be an identical LKB1 null background); in fact there is wide variability with about 6 fold growth in the R87K experiment (Fig 6C) and more than 20 fold growth in the Y49k experiment. This suggests that there could be some leakiness in the dox promoter or that there are simply differences in growth rates across the different stable cell lines. Therefore basing comparisons in the growth with or without dox when the baseline is different for each does not seem reasonable. It would be more appropriate to see if similar mutation-specific differences are observed across different models expressing the particular mutant.

We thank the reviewer comment. Conceptually, all growth rates changes are referred to the parental cells. Then, the observed effect after expression of every different isoform against to their parental cells is compared against the effect observed in the wild type isoform. While it might be true that the growth rates across the different infected cell lines is different, there are other issues that might influence the basal growth rate in the mouse (such as: survival of cells after injection within the mouse)

that are difficult to control from mouse to mouse experiment. Another example of variability is when injecting same cell line and same number of cells in different mice, tumor engraftment occurs at different time-points. In the case of Y49D mutant the expression of the mutant increased tumor growth even if the basal growth was higher than the other experiment. We think that this does not invalid the result showing that expression of this mutant increase tumor growth. We agree with the reviewer saying that it would be good to investigate whether similar mutation-specific differences are observed across different models expressing the particular mutant, but that is out the scope of the study, and considering tumor heterogeneity (different genetic tumor-cell background), obtaining different results, would not invalid these ones.

8. LKB1 is a master kinase that activates several downstream pathways by phosphorylating different substrates. In term of characterization of loss of function and kinase activity in the different mutant isoforms, data provided in the manuscript should be further supported by assessing LKB1 downstream targets such us AMPK/mTOR (cell proliferation and metabolism control) and/or MARK/SAD/SIK (cell polarization) should be performed. Recent papers (Hollstein et al, Cancer Discovery 2019 and Murray et al Cancer Discovery 2019) are highly relevant in this regard and should be cited.

We appreciate the reviewer suggestion. To confirm the described kinase activity of the different LKB1 isoforms we have investigated the response to metabolic stress (glucose starvation) having as a readout the surrogate marker p-AMPK confirming the observed results. We have generated a new Figure 2D panel and old figure 2D has become Figure 2E.

9. In vitro assays in figure 1, as well as, in vivo experiments shown in figure 6 revealed a potentially interesting result: expression of the G135R and D194Y mutations in an LKB1 deficient cell line (A549), accelerated tumor growth compared with parental cells, which already has a high proliferative phenotype due to lack of LKB1 expression. This would suggest that the mutations could activate pathways beyond simply LKB1 loss. This should be evaluated in a different model and the authors should discuss possible explanations for this phenomenon.

This is a good observation that we pursuing separately. It is very interesting how a tumor suppressor conceptually based on the presence or absence of the protein, could acquire oncogene features upon punctual mutations. This is supported by the case of D194 position which is the most frequent selected missense mutation across different type of cancers. In supplementary Figure 1C it is shown that this effect is also observed in the colony formation assay performed in two additional models, G361 melanoma cells and HeLa cells. In addition to this, we have generated proliferation

data in G361 melanoma cells showing the same effect (showed below). We are currently investigating this subject.

The finding that some mutants may accelerate tumor growth (e.g. Figure 6D) compared with the null background does not seem to fit with the earlier in vitro experiments showing that the mutants did not suppress tumor growth but did not accelerate it either (figure 1), which would be more consistent with the role of a tumor suppressor. The statement on page 16 “Thus, the in vitro tumor suppressor capabilities of the investigated LKB1 mutants were reflected in vivo.” is not accurate; there is no significant suppression for the R87K and there is significant acceleration in Fig 6B, D, and E. The authors should discuss the differences in these findings.

We thank the reviewer comment. We wanted to clarify that the experiments that the reviewer is referring to, are not equivalent and the results although in the same direction might differ in intensity due to multiple factors. First, and probably more important is the -in vitro, in vivo- feature of the experiments. Cells in tissue culture media are not exposed to the same factors conditions than in vivo that might determine the final result. In addition to this, in "in vivo" settings, there is always a selection of what will grow better within the injected cell lines (cell lines are heterogeneous). Moreover, the timing for exponential multiplication is different when we talk about in vitro or in vivo experiments. We believe that the effect that we described in vitro is reflected in vivo. In the case R87K where the reviewer states that the suppression is not significant. We think that there is a suppression, might be not significant for different reasons but there is a clear tendency of tumor suppression in vivo. In other words, with the results observed in Figure 6 we could not say R87K mutant either lost its tumor suppressor capabilities nor promoted tumor growth. In the case of Y49D, the observed effect in vitro is that this mutant has lost its tumor suppressor capabilities (96 hours, 5-7 days in the colony formation). In the in vivo experiment up to day 27 even in day 30 the observed effect is exactly the same, this mutant has lost its tumor suppressor capabilities. Is in the final 5-7 days when tumors took off, which might be cause of the swelling (inflammation). In the case of G135R, expression of this mutant promotes cell proliferation (96h in vitro). This effect was much more pronounce in vivo

probably because a number of factors including cell selection and richer environment, but the effect was the same, it promoted growth beyond the tumor suppressor function. Same applies for D194Y mutant, so we believe that, with proper cautious, there are not such differences and the observed effect described in vitro for every mutant was reflected in vivo. We hope this explanation satisfy the reviewer concerns.

10. In figure 1, data revealed that Q135R and D194Y mutations increased cell proliferation when expressed in a LKB1 deficient cell line (A549), while R87K and wt isoforms reduced cell growth. By contrast, Y49D mutant isoform seems not to have effects in term of cells proliferation. In figure 2, by contrast, Q135R and D194Y only partially lost kinase activity, while Y49D also significantly reduced its kinase activity and this isoform seems to lose ability to interact with STRAD α . On the other hand, R87K (wt-like isoform), Y49D and D194K, but not G135R display a shorter protein half-life. These data are inconsistent with the proliferative phenotype shown in figure 1. The authors should discuss possible explanations for these findings.

We appreciate the reviewer observation, but we do not understand which is the inconsistent link between the half-life of the proteins, the relative kinase activity of every mutant and proliferation.

As far as we know, tumor suppressor capabilities are still working in LKB1 molecules force to be cytoplasmic (w/o nuclear localization signal) having a kinase activity (Tianien te al., 2002). D194Y and G135R are nuclear and cytoplasmic but they have minimized their kinase activity this is translated into a lack of tumor suppression activity. In the case of Y49D not only does not have a full kinase activity, but it is nuclear. These features are promoted by the lack of binding of Y49D to STRAD α . Consequently, this mutant has lost its tumor suppressor capabilities. In the case of R87K, its kinase activity and localization is indistinguishable from the wild-type isoform, this why it conserve the tumor suppressor capabilities. In relation to the link between proteins stability and proliferation, we are sorry but, we do not understand what is the reviewer referring to. We hope this explanation satisfy her/his concerns.

Minor

issues:

1. The statement (page 20) that “LKB1 Y49D showed signs of inflammation and disorganized angiogenesis (hemorrhagic) confirming the role of LKB1 in regulating cytokine production and inflammation” is an overstatement because the models were tested in immunocompromised mice so inflammation could not be accurately assessed; furthermore, hemorrhage is not an established marker for inflammation.

The reviewer is right, this statement was not very accurate, we have changed the term inflammation for inflammation-like, and corrected the whole sentence accordingly. In the case of hemorrhage, we did not mean to say that was a marker of inflammation, we just wanted to link two different events that have been linked in the literature in both

directions Hemorrhage promotes inflammation (Ahn SH et al., 2019) and inflammation induces hemorrhage (Valance Washington A. et al 2009; Goerge et al., 2005).

2. In Fig 1E: panel with D194Y at 48h + Dox has dark blue bar (G1 arrest) in the middle of the other two blue bars, whereas elsewhere the dark blue bar is at the bottom. Should the bar be dark blue (meaning G1 arrest), in which case it should be at the bottom, or medium blue? Also, elsewhere in the figure, there are 4 different shades of blue but only 3 in the legend- one color is nearly black. Please clarify the colors and labels.

We believe that the reviewer refers to Figure 1F. The color labels for G1, S and G2 phases of the cell cycle are indicated in the upper part of the graphs (three different blue colors). As stated in the figure legend: "Dark blue bars represent cell cycle phases showing significant changes (n=3, p<0.05 calculated by Student's t-test)". We are sorry for not being more explicit.

3. The concluding statement in the Discussion is not adequately supported by the data, particularly the effects on immune modulation given the immunocompromised mouse models: "In summary, we show that beyond the role of the non-mutated protein as a tumor suppressor, missense LKB1 somatic mutations could contribute to tumor development and/or progression by modifying not only intrinsic cell capabilities such as proliferation, motility or adhesion but also the tumor microenvironment, affecting inflammatory responses and likely the immune system. "These experiments could be repeated in syngeneic murine models to better get at the impact on the microenvironment. In addition, public data could be mined (e.g. TCGA using CIBERSORT) to see if it supports the association of different patterns associated with different mutations.

We appreciate the reviewer comment and we agree in that our model does not reflect the complex and orchestrated anti-tumoral immune response that would happen in an immunocompetent mouse model. However, one of the main purposes of this manuscript is to emphasize the possible roles of mutated LKB1 molecules in cancer. Results from different animal models in our lab (LKB1 conditional knocked out in different tumor models (lung and skin)) suggest a role of LKB1 in inflammation. Furthermore, it has been demonstrated that the pro-inflammatory effects of germline deletion mutations in the tumor suppressor gene STK11 on immune T cells lead to cancer predisposition syndrome Peutz-Jeghers syndrome (PJS) in mice (Poffenberger et al., 2018). Additionally, this type of data is also reflected on the regulation of some of the molecules involved in inflammation (CCL17, MMP2, TIMP3, LIF, ILR2,... etc) in the studies investigating the RNA expression profiles of tumors lacking LKB1 expression vs. LKB1 expressing tumors (Kaufman et al., 2014). Thus, even though we agree that the used mouse model is not appropriate to study the mechanisms involved in anti-tumoral immune responses. The data from the expression profiles (the nature of

some of the regulated molecules), the RT-PCR results (differential regulation of cytokines by LKB1 isoforms), the secretome analysis, together with some of the phenotypes observed in the tumors (expression of IL8, GRO α , β and γ) support the role of some STK11 somatic mutations in modifying the tumor microenvironment and immune-related processes. Our intention is not to explain the particular mechanism/s involved in the immune response regulation, but to evidence the different biological processes affected by the missense mutation of a tumor suppressor usually lost in tumors.

4. Analysis of cells cycle phase distribution is shown in Figure 1F to support higher or lower rate of proliferation across the different mutants. Some issue with these data:
- First, quantification of all cell cycle phases together should be close to 100%. Are the authors missing any population? For example, is SubG1 peak (Dead cells which display lower probe staining) quantified? Please, reanalyzed carefully these data.

The reviewer is right. The cause for this was that in the original figure we only considered the 2n population. These cells have a polyploid population that oscillates and we did not include in the data (See figure below). We have corrected this in the manuscript and this time the data is referred to the 100% of the 2n population.

- Cells cycle analysis are typically performed when cells are growing exponentially to avoid include additional effects that could interfere in the cell cycle progression. Could the authors explain why G1 phase percentage is higher in -Dox treatment at 48h compared with 0h for WT and Y49D isoform? If cell cycle analysis was performed when cells are growing exponentially, G1 phase percentage at 48h should be similar to 0h. This data seems to indicate a G1 arrest at 48h due to low nutrient availability or confluent state rather than expression of LKB1 wt. Initial number of plated cells should be recalculated to allow exponential grow still at 48h.

As above explained this figure has been corrected accordingly.

- Finally, averages for 2 or 3 independent experiments should be included.

As explained above and in the figure legend these experiments were done in triplicates where dark blue bars represent significative differences

5. In figure 1D and Figure S1 C authors show the quantification of clonogenic assay performed in the different mutated isoform. Are these differences statistically difference? If it does, please include this data.

We apologize for not include this in the original version of the manuscript. This issue has been corrected accordingly.

6. Figure 6C is cited in the text before than Figure 6B.

We apologize for this mistake. This issue has been fixed.

7. In Discussion section, second paragraph the sentence “Thus, selected missense LKB1 selected mutations...” should be corrected by deleting the second “selected”.

We have corrected this mistake.

Reviewer #3 (Remarks to the Author):

This manuscript by Paula Granado-Martinez et al. focuses on the functional study of STK11 (LKB1) missense somatic mutant isoforms such as LKB1Y49D, LKB1R87K, LKB1G135R and LKB1D194Y in cancer. They performed several experiments including omics analysis such as secretome to reveal the functions of these mutant isoforms. This is interesting work to reveal the importance of the STK11 (LKB1) missense somatic mutant isoforms. Before it can be published in Communications Biology, I have several suggestions as described below.

1. The statistics analysis should be performed in Figure 1D, 3A and 3E.

We thank the reviewer comment. We have added the statistics analysis to the suggested figures.

2. The supplementary tables for proteome should be described the protein full names and the detail information about mass identification. Since many errors happened in protein identification even using software such Proteome Discoverer, to check the mass peaks carefully and list the results are important in proteome field.

We apologize form not include this information in the previous version. This information has been added to the new version of the manuscript as Supplementary Table 2.

3. I strongly suggest that the authors should deposit the proteome data to public database such as ProteomeXchange (<http://www.proteomexchange.org/>).

We appreciate and agree with the reviewer suggestion. We have deposited the data in proteomeXchange database PRIDE with accession number: PXD018041.

List of new data added to manuscript

- 1- We have added in the introduction an explanatory phrase justifying the selection of the mutants (page 5 lines 7-9).
- 2- We have added to supplementary Figure S1 a western blot showing the expression amounts of endogenous LKB1 in 14 lung tumor cell lines and 8 melanoma cell lines.
- 3- We have added a new panel Figure 1E showing the metabolic profiles (Seahorse technology: OCR, ECAR, and some mitochondrial parameters for mitochondrial use of the different cell lines expressing the different isoforms of LKB1.
- 4- In supplemental Figure S1 we have added a new panel E showing the IC50 for metformin of all cells expressing the different isoforms of LKB1.
- 5- We have compared our gene expression dataset with all the studies in Kaufman et al 2014, individually and collectively, confirming and validating the genes presented and analyzed in our study. A new panel in Supplementary Figure S3A has been added.
- 6- We added the comparative regulation of three more genes (PLAUR, CYP1B1 and DUSP1) among cell lines expressing the different LKB1 isoforms to the Figure 4D. These genes were regulated in both our dataset and in the datasets investigated in Kaufman et al., 2014.
- 7- We have added a new panel in figure 2D showing a physiological functional assay to confirm the described kinase activity. We have measured the response of cells to metabolic stress measuring the amounts of p-AMPK as a surrogate marker of the LKB1 kinase activity.
- 8- We have fixed the cell cycle experiment following the reviewer suggestions.
- 9- We also have added all the statistical analysis to the graphs lacking the p-values.
- 10- We have added new Supplementary Table 2 showing the proteins detected by mass spectrometry in secretomes before and after the expression of the different LKB1 isoform. Old Supplementary Table 2 become Supplementary Table 3 and Supplementary Table 3 has become Supplementary Table 4.
- 11- We have uploaded the gene expression data and the proteomics data to Arrayexpress and ProteomeXchange databases.

Reviewers' comments:

Reviewer #1 (Remarks to the Author):

The Authors revised the manuscript according to my suggestions and in my opinion the quality of the paper is now clearly improved. However, it still needs minor revision as follows:

1. More than one reviewer asked about the rationale of selecting these 4 STK11 mutations and the Authors' reply is convincing. However, I encourage them to include in the paper (either as panel of a main figure or as supplementary figure) the nice dot plot included in their rebuttal letter and showing the position of STK11 mutations reported in tumors (including the four mutations functionally characterized here).
2. Figure 1, panel g. Please indicate in the Seahorse analysis graphs the time points when the various inhibitors (oligomycin, FCCP, Rotenone) were added to the wells. Although this is obvious to the expert in the field it might help understand the graph to other readers.
3. In the M&M section, chapter Gene expression analysis, the Authors state that genes were considered differentially expressed in A549 cells if the fold change was >1.5 and the p was <0.05 . However, in Figure legend 4a they state that GSEA used cut-off values >1.2 -fold. Why were two different cut-off used?
4. Figure 4b, expressing should read expressing. Please fix the typo.
5. Discussion: transcriptome analysis indicates that all four STK11 mutants showed dis-regulation of genes involved in intracellular trafficking of vesicles and endosomes, including AP1S3 and RUSC2. Which are the functional implications of these observations? Is vesicles trafficking predicted to be compromised in cells bearing these STK11 mutations? Could the Authors briefly speculate on this?

Reviewer #2 (Remarks to the Author):

The authors have revised the manuscript and addressed many specific concerns from the earlier review. There remain, however, several concerns that remain.

1. Perhaps the most potentially interesting finding of the study is that different mutations may have different functional implications; for example, if a particular STK11 mutation were found to have higher levels of a particular cytokine secreted, it is possible a therapeutic strategy could be developed for it. Unfortunately, much of the data regarding the differences between the mutation rests with gene expression or proteomic data on A549 expressing different mutations or a limited number of other models. Given the large number of genes that are differentially regulated by mutation (e.g. Figure 4D), there is a significant risk for false positives and it is always possible to construct biologically plausible pathways with that many genes (e.g. the pathway maps in Figure 5) but the significance of any of these pathways and their association with a specific mutation remains unclear. If the authors would like to demonstrate that specific mutations lead to differences in pathway activation, it would be more convincing to show that changes observed in vitro with a specific mutation (e.g. D194) can be validated using tumors from patients with that specific mutation (e.g. using TCGA data or Kaufman dataset). Alternatively, it would be useful to validate that expression of specific mutations lead to the same differences in an independent in vitro system (e.g. different cell line).
2. The authors state (page 15) "To validate the relevance of our dataset, we compared our 817

regulated genes with the 2080 unique regulated genes obtained from 15 different datasets (top 200 regulated genes in each data set) published in Kaufman et al., 201437, comparing the gene expression profiles of human and murine tumors with or without deleted STK11". It is unclear how this is validating the relevance of the dataset. While some overlap in the genes differentially regulated in STK11 mutant vs wt was observed with the current dataset, it does not seem that the differences induced by specific mutations was validated- e.g. the analyses of genes differentially regulated by D194 mutants vs other STK11 mutants in the Kaufman dataset compared with the genes or proteins specifically regulated by D194 in the current study.

3. In the secretome analysis, how many of the proteins that were differentially expressed in a mutation-specific manner were also seen to be different expressed at the RNA level from the transcriptomic analysis? This should be included.

4. The authors continue to note differences in the pattern of angiogenesis and draw conclusions from this: "Remarkably, all tumors expressing LKB1Y49D showed signs of inflammation- like and disorganized angiogenesis (hemorrhagic), confirming the role of LKB1Y49D in regulating cytokine production and inflammation related processes."

- i) These changes should be quantitated and shown to differ between the different xenografts.
- ii) Furthermore, as noted in the earlier review, this is overinterpreting the data from a subcutaneous tumor model, and there are many other explanations for hemorrhagic vasculature (e.g. differences in pericyte coverage as seen with the Ang/Tie2 system; changes in hypoxia; etc), and there are no mechanistic experiments to link changes observed with changes in the vasculature or inflammation.
- iii) While inflammation may be linked to vascular pattern, there is no evidence it is linked here and the authors should not draw conclusions about inflammation without examining inflammatory cells (hard in an immunocompromised model). It would be reasonable to make statements about specific cytokines if evidence supports it.

Minor issues:

1. Figure 1G (and elsewhere): parental or empty vector control should be included so that the degree of alteration in ECAR in the absence of LKB1 vs with expression of wt LKB1 or mutants can be assessed.
2. Figure 1, panel g. Please indicate in the Seahorse analysis graphs the time points when the various inhibitors (oligomycin, FCCP, Rotenone) were added to the wells. Although this is obvious to the expert in the field it might help understand the graph to other readers.
3. In the M&M section, chapter Gene expression analysis, the Authors state that genes were considered differentially expressed in A549 cells if the fold change was >1.5 and the p was <0.05 . However, in Figure legend 4a they state that GSEA used cut-off values >1.2 -fold. Why were two different cut-off used?
4. Figure 4b, expresing should read expressing. Please fix the typo.

Answer to reviewers in a point by point basis

Reviewers' comments:

Reviewer #1 (Remarks to the Author):

The Authors revised the manuscript according to my suggestions and in my opinion the quality of the paper is now clearly improved. However, it still needs minor revision as follows:

1. More than one reviewer asked about the rationale of selecting these 4 STK11 mutations and the Authors' reply is convincing. However, I encourage them to include in the paper (either as panel of a main figure or as supplementary figure) the nice dot plot included in their rebuttal letter and showing the position of STK11 mutations reported in tumors (including the four mutations functionally characterized here).

Following the reviewer suggestion, we have added this figure as Supplementary Fig. 1a. and cited accordingly to the new nomenclature the rest of the figure panels in the text.

2. Figure 1, panel g. Please indicate in the Seahorse analysis graphs the time points when the various inhibitors (oligomycin, FCCP, Rotenone) were added to the wells. Although this is obvious to the expert in the field it might help understand the graph to other readers.

We agree with this reviewer and have corrected accordingly in the graphs of Fig. 1g

3. In the M&M section, chapter Gene expression analysis, the Authors state that genes were considered differentially expressed in A549 cells if the fold change was >1.5 and the p was <0.05. However, in Figure legend 4a they state that GSEA used cut-off values >1.2-fold. Why were two different cut-off used?

We apologize for this mistake. The corresponding section of the M&M has been corrected accordingly.

4. Figure 4b, expresing should read expressing. Please fix the typo.

This typo has been corrected.

5. Discussion: transcriptome analysis indicates that all four STK11 mutants showed dis-regulation of genes involved in intracellular trafficking of vesicles and endosomes, including AP1S3 and RUSC2. Which are the functional implications of these observations? Is vesicles trafficking predicted to be compromised in cells bearing these STK11 mutations? Could the Authors briefly speculate on this?

Following the reviewer suggestion, we have added a paragraph in the discussion section (page 14) speculating the possible effects of the dysregulation of these genes.

Reviewer #2 (Remarks to the Author):

The authors have revised the manuscript and addressed many specific concerns from the earlier review. There remain, however, several concerns that remain.

1. Perhaps the most potentially interesting finding of the study is that different mutations may have different functional implications; for example, if a particular STK11 mutation were found to have higher levels of a particular cytokine secreted, it is possible a therapeutic strategy could be developed for it. Unfortunately, much of the data regarding the differences between the mutation rests with gene expression or proteomic data on A549 expressing different mutations or a limited number of other models. Given the large number of genes that are differentially regulated by mutation (e.g. Figure 4D), there is a significant risk for false positives and it is always possible to construct biologically plausible

pathways with that many genes (e.g. the pathway maps in Figure 5) but the significance of any of these pathways and their association with a specific mutation remains unclear. If the authors would like to demonstrate that specific mutations lead to differences in pathway activation, it would be more convincing to show that changes observed *in vitro* with a specific mutation (e.g. D194) can be validated using tumors from patients with that specific mutation (e.g. using TCGA data or Kaufman dataset). Alternatively, it would be useful to validate that expression of specific mutations lead to the same differences in an independent *in vitro* system (e.g. different cell line).

We agree that validating our results in other models will be ideal, and probably some of our results will be more universal than others. Cancer is a very heterogeneous disease (even intratumoral) and LKB1 is a multitask protein. Contributing somatic mutations are selected through tumor evolution according to the mutational context and the timely tumor needs. Due to the nature of LKB1, mutations in this molecule might affect cells in different ways according to their mutational context and molecular rewiring of the cell. There are already examples showing that different tumor types harboring a particular mutation (i.e. a BRAF^{V600E}) behave differently, promoting different scenarios and therapeutic responses (i.e. melanoma and thyroid or colon cancer). We believe that our results are solid and strong. We obtained the data using different technical approaches, from a global view (OMICs), to the detail (biochemical and molecular techniques), *in vitro* and *in vivo*. It is possible that validation of our results could make the described mechanisms broader, but not with better quality nor relevant. We could validate our results in different systems and still have the same results, but we can also obtain different results and that would not invalidate the first ones. Obviously, there are some results within the manuscript that are hypothesis-generating, and of course, all of the results will require independent validation in future work. In this particular case, using human samples to validate our results is almost impossible due to the low number of samples with these particular mutations and the limited information available about them (i.e. RNA seq, Exome...). The manuscript that the reviewer refers to (Kaufman et al 2014) is dedicated to study the “loss of function of LKB1” vs. samples expressing LKB1 (at least at the mRNA level), not mutations. If we analyze the more frequent mutation (D194 residue) in the Lung adenocarcinoma Pan Cancer TCGA dataset (503 samples, included in Kaufman et al 2014) there are 3 samples mutated in D194 residue. We have analyzed the mRNA expression levels of STK11 vs. mRNA levels of the genes expressed differentially upon the expression of STK11 isoforms (WT vs. mutants, specially D194Y). As shown in the below figure the expression of the analyzed genes in the D194 samples are mostly in agreement with our results (Fig.4). However, the interpretation of these results with just 3 samples, no control of the protein amounts of LKB1 within samples, unknowing the mutation allele frequency of STK11 mutant in tumor cells, heterogeneity of tumor expressing this allele, tumor cell percentage of the sample... etc., it turns difficult. The real validation of this type of data will come from the rest of the scientific community over time, and the Kaufman study is a good example, where 15 independent studies investigated the same thing over time, in different groups of samples, tumor subtypes, and species.

Lung Adenocarcinoma (TCGA, PanCancer Atlas) Complete samples (503 patients/samples)

2. The authors state (page 15) “To validate the relevance of our dataset, we compared our 817 regulated genes with the 2080 unique regulated genes obtained from 15 different datasets (top 200 regulated genes in each data set) published in Kaufman et al., 2014³⁷, comparing the gene expression profiles of human and murine tumors with or without deleted STK11”. It is unclear how this is validating the relevance of the dataset. While some overlap in the genes differentially regulated in STK11 mutant vs wt was observed with the current dataset, it does not seem that the differences induced by specific mutations was validated- e.g. the analyses of genes differentially regulated by D194 mutants vs other STK11 mutants in the Kaufman dataset compared with the genes or proteins specifically regulated by D194 in the current study.

In the first revision round the reviewer suggested the comparison of our regulated genes with public datasets (Kaufman et al., 2014) to “A) focus on genes known to be regulated by LKB1 in clinical datasets, and then B) Assess whether those particular LKB1-regulated genes are differentially

regulated by different mutants. We found this suggestion a very good contribution to assure that the selected genes were specific for LKB1 so we could compare the regulation of these genes by the LKB1WT with the mutant isoforms. It turned out that all the genes we had selected were also relevant in the manuscript (Kaufman et al., 2014, this information was included in the revised version of the manuscript). We believe that this suggestion strengthened our results and validated the selection of the genes regulated by the expression of LKB1 WT that was analyzed with the LKB1 mutant isoforms. As above explained due to the restricted number of samples harboring the appropriate mutations and data availability of these samples the type of analysis suggested are not significant (Figure enclosed).

3. In the secretome analysis, how many of the proteins that were differentially expressed in a mutation-specific manner were also seen to be different expressed at the RNA level from the transcriptomic analysis? This should be included.

This is something very interesting but we cannot answer it. We never did gene-array expression profile of the mutant isoforms. We just did it for the WT isoform and this comparison was included in supplementary figure S3C (1st version of the manuscript).

4. The authors continue to note differences in the pattern of angiogenesis and draw conclusions from this: "Remarkably, all tumors expressing LKB1Y49D showed signs of inflammation- like and disorganized angiogenesis (hemorrhagic), confirming the role of LKB1Y49D in regulating cytokine production and inflammation related processes."

- i) These changes should be quantitated and shown to differ between the different xenografts.
- ii) Furthermore, as noted in the earlier review, this is overinterpreting the data from a subcutaneous tumor model, and there are many other explanations for hemorrhagic vasculature (e.g. differences in pericyte coverage as seen with the Ang/Tie2 system; changes in hypoxia; etc), and there are no mechanistic experiments to link changes observed with changes in the vasculature or inflammation.
- iii) While inflammation may be linked to vascular pattern, there is no evidence it is linked here and the authors should not draw conclusions about inflammation without examining inflammatory cells (hard in an immunocompromised model). It would be reasonable to make statements about specific cytokines if evidence supports it.

Following reviewer suggestions, we have softened our message and substituted the sentence "*Remarkably, all tumors expressing LKB1Y49D showed signs of inflammation- like and disorganized angiogenesis (hemorrhagic), confirming the role of LKB1Y49D in regulating cytokine production and inflammation related processes.*" By *Interestingly, all tumors expressing LKB1^{Y49D} showed signs of swelling supporting a role of LKB1^{Y49D} in regulating cytokine production and inflammation-related processes*".

Minor

issues:

1. Figure 1G (and elsewhere): parental or empty vector control should be included so that the degree of alteration in ECAR in the absence of LKB1 vs with expression of wt LKB1 or mutants can be assessed.

The experiment related to the specific question raised by reviewer #1 in the previous revision was planned according to answer the particular question formulated by the reviewer. In fact the reviewer #1 was satisfied with that answer. The experiments the reviewer is asking is out of the scope of this study.

2. Figure 1, panel g. Please indicate in the Seahorse analysis graphs the time points when the various inhibitors (oligomycin, FCCP, Rotenone) were added to the wells. Although this is obvious to the expert in the field it might help understand the graph to other readers.

Following the reviewer suggestion, we have added this figure as Supplementary Fig. 1a. and cited accordingly to the new nomenclature the rest of the figure panels in the text.

3. In the M&M section, chapter Gene expression analysis, the Authors state that genes were considered differentially expressed in A549 cells if the fold change was >1.5 and the p was <0.05 . However, in Figure legend 4a they state that GSEA used cut-off values >1.2 -fold. Why were two different cut-off used?

We apologize for this mistake. The corresponding section of the M&M has been corrected accordingly.

4. Figure 4b, expresing should read expressing. Please fix the typo.

This typo has been corrected.